# NOVA1 regulates *hTERT* splicing and cell growth in non-small cell lung cancer

Andrew T. Ludlow [1,2], Mandy Sze Wong[1,3], Jerome D. Robin[1,4], Kimberly Batten[1], Laura Yuan[1], Tsung-Po Lai[1], Nicole Dahlson[1], Lu Zhang[1], Ilgen Mender[1], Enzo Tedone[1], Mohammed E. Sayed[1,2], Woodring E. Wright [1] & Jerry W. Shay[1]

Alternative splicing is dysregulated in cancer and the reactivation of telomerase involves the splicing of *TERT* transcripts to produce full-length (FL) *TERT*. Knowledge about the splicing factors that enhance or silence FL *hTERT* is lacking. We identified splicing factors that reduced telomerase activity and shortened telomeres using a siRNA minigene reporter screen and a lung cancer cell bioinformatics approach. A lead candidate, NOVA1, when knocked down resulted in a shift in *hTERT* splicing to non-catalytic isoforms, reduced telomerase activity, and progressive telomere shortening. NOVA1 knockdown also significantly altered cancer cell growth in vitro and in xenografts. Genome engineering experiments reveal that NOVA1 promotes the inclusion of exons in the reverse transcriptase domain of *hTERT* resulting in the production of FL *hTERT* transcripts. Utilizing *hTERT* splicing as a model splicing event in cancer may provide new insights into potentially targetable dysregulated splicing factors in cancer.

[1] Department of Cell Biology, UT Southwestern Medical Center, 5323 Harry Hines Boulevard, Dallas, TX 75390, USA. [2] School of Kinesiology, University of Michigan, 401 Washtenaw Ave., Ann Arbor, MI 48109, USA. [3] Present address: Cold Spring Harbor Laboratories, One Bungtown Road, Cold Spring Harbor, New York, NY 11724, USA. [4] Present address: Aix-Marseille University, Marseille Medical Genetics (MMG), UMR125, Marseille 13385, France. Correspondence and requests for materials should be addressed to A.T.L. (email: atludlow@umich.edu)

Telomerase is a tightly regulated ribonucleoprotein complex (RNP) that maintains or lengthens human telomeres by adding 5′-TTAGGG repeats. Telomerase consists minimally of a reverse transcriptase (RT) protein catalytic subunit (hTERT) and a template RNA, telomerase RNA (hTERC[1]). Embryonic and transit amplifying stem cells have telomerase activity while most somatic cells do not. However, 85–90% of cancers utilize telomerase activity to maintain telomeres[2]. In most human cancers, the limiting factor in telomerase activity is the expression level of the RT, hTERT. hTERT is regulated by transcriptional and post-transcriptional mechanisms[3–5]. Transcriptional regulation of hTERT has been extensively studied, however the findings do not completely explain how telomerase is regulated in cancer. How the transcribed messenger RNA is processed (i.e., RNA-processing events), which is critical for determining if active telomerase is produced or not, is less well understood. One such RNA-processing regulatory mechanism is alternative splicing, which contributes to protein diversity and transcript abundance[6]. hTERT produces a transcript containing 16 exons that can be spliced into multiple isoforms[7–10], including the full-length (FL) RT competent form. In tumor cells and dividing stem cells, FL hTERT and several spliced variants are co-expressed at detectable levels. Since telomerase activity is almost universally activated in human cancer, further research into the mechanisms that regulate hTERT mRNA processing, specifically alternative splicing, may provide additional clues about telomerase regulation in cancer and importantly could elucidate new candidate genes to target for telomerase inhibition and for anticancer therapies.

Of the hTERT splice isoforms, the four major isoforms that have been studied involve exons 5–9, which encode the RT domain of hTERT (Supplementary Table 1, Supplementary Figure 6). The four major isoforms are a result of splicing of regions termed "alpha" and "beta" located within exons 5–9[10]. Only the "FL" version (α+β+; FL) containing all five intact exons of the RT domain has the potential to encode catalytically active enzyme[8,10,11]. The other isoforms are generated by skipping of exons 7 and 8 (α+β−; minus beta), which introduces a frameshift and premature stop codon in exon 10, the skipping of 36 nucleotides (nts) of exon 6 (α−β+; minus alpha), which is in frame and generates a dominant-negative RT incompetent telomerase[10], and (α−β−; minus alpha-beta), which has both skipping events ([10]; Supplementary Table 1). Other variants of hTERT exist that result from splicing events outside of the RT domain[12] (Supplementary Table 1). The important regulatory sequences and splicing factors that bind hTERT pre-mRNAs to produce the RT competent versus RT-deficient hTERT splice isoforms are not well described.

Very few investigations into the cis- and trans-acting factors that regulate splicing of hTERT have been performed. Our group previously identified highly conserved sequences in old world primates, including humans, that regulate hTERT splicing choice[13,14]. However, little is currently known about the trans-acting factors that bind these regulatory regions. Thus, identification of such proteins would close a significant gap in telomerase regulation knowledge and also potentially identify protein targets to shift the splicing of hTERT message to inactive forms to reduce telomerase activity, progressively shorten telomeres, and ultimately leading to reduced tumor growth in vivo. RNA-binding proteins target multiple genes; thus, it is likely that identification of a protein that targets hTERT may impact other important pathways that are cancer cell dependencies.

Recently, three splicing proteins, SRSF11, hnRNPL, and hnRNPH2, when overexpressed in cancer cells were identified to potentially regulate hTERT minus beta splicing choice using an hTERT minigene[15]. There are more than 500 RNA-binding proteins encoded in the genome and splicing is the result of cellular context, RNA secondary structure, RNA editing, and competition for splice sites, therefore much is left to be learned concerning hTERT splicing regulation[16–18].

To address the protein networks that regulate the alternative splicing of FL hTERT in cancer cells, we took two approaches: a hTERT dual-luciferase minigene splicing reporter RNAi screen and a bioinformatic analysis of a panel of highly characterized human lung cancer cell lines to identify genes that may regulate hTERT splicing and other cancer cell phenotypes. These two approaches identified neuro-oncological ventral antigen 1 (NOVA1) as a candidate gene. In non-small cell lung cancer cells that express high levels of NOVA1, we found that stable reduction in NOVA1 levels shifted hTERT splicing toward inactive transcripts, reduced telomerase activity, which led to progressively shortened telomeres. We also demonstrated that NOVA1 knockdown reduced migration through extracellular matrices, and resulted in smaller tumors in vivo. Thus, the experiments described in the present report provide a mechanistic view of how cancer cells regulate hTERT splicing.

## Results

### hTERT minigene small interfering RNA screen of RNA-binding proteins.
To investigate the protein factors involved in alternative splicing of hTERT we performed a small interfering RNA (siRNA) screen in HeLa cells stably expressing an hTERT minigene splicing reporter (Fig. 1). The minigene used in this study allowed for luciferase-based measurements of hTERT "FL" (intact exons 6–8) and minus beta hTERT (skipping of exons 7 and 8). Fluc indicates minus beta splicing (inactive hTERT) and FL hTERT is indicated by Rluc (Fig. 1a).

We obtained pools of four siRNAs to 516 RNA-binding protein genes. HeLa cells containing the minigene reporter were transfected with the siRNAs and luciferase measurements were made 72 h later (Fig. 1a). The screen repeat runs were highly correlated ($R^2 = 0.794$, Pearson's r, Supplementary Figure 1C). Of the 528 siRNAs tested, 97 siRNAs targeting 93 unique genes resulted in a twofold or greater increase in minus beta splicing and 20 siRNAs targeting 17 genes resulted in a twofold or greater increase in FL splicing (a list of these gene identifiers can be found in Supplementary Data file 1). Our objective was to find targets that had cancer specificity (i.e., not widely or highly expressed across normal adult human tissues) for potential follow-up studies; thus, we focused our follow-up studies on the 93 genes that when knocked down resulted in twofold or greater increases in the ratio of minus beta to FL. We pursued a parallel bioinformatics approach to narrow down the list of candidate genes using a highly characterized panel of human non-small cell lung cancer cell lines.

### Bioinformatics analysis of 516 RNA-binding proteins.
We measured telomere biology phenotypes (telomerase activity, hTERT expression, and telomere length[19]) in 17 well-characterized lung cancer cell lines and correlated expression of splicing factors and telomere biology between these cell lines to narrow down our target gene list (Supplementary Figure 1D, Supplementary Figure 1E). We found that telomere length and telomerase activity were significantly correlated ($R^2 = 0.51$, $p = 0.001$, Pearson's r, Supplementary Figure 1G). hTERT mRNA expression of exons 7/8 and exons 15/16 showed a modest but significant correlation with telomere length ($R^2 = 0.38$, $p = 0.008$ for exons 7/8 and $R^2 = 0.30$, $p = 0.02$ for exons 15/16, Pearson's r; Supplementary Figure 1H). Telomerase activity correlated with exons 7/8 expression of hTERT ($R^2 = 0.27$, $p = 0.03$, Pearson's r; Supplementary Figure 1I). We quantified hTERT splice isoforms

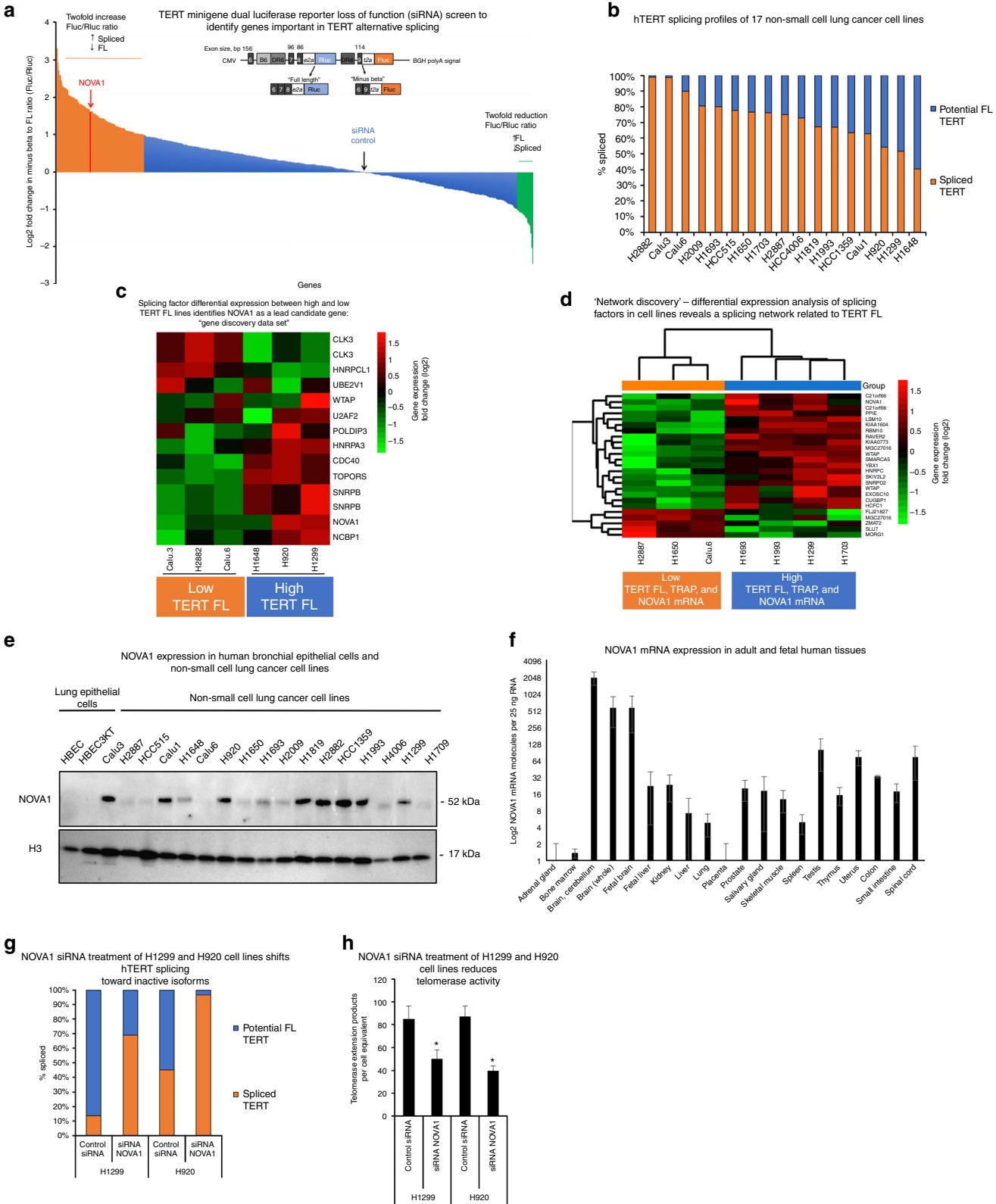

(Fig. 1b, primer sequences are in Supplementary Table 2), we choose to analyze cell lines that differed significantly by the percentage of FL *hTERT* expression, percentage of minus beta *hTERT* expression, and telomerase activity (three lines at each extreme, Fig. 1c, "Gene Discovery Data set"). We correlated the expression data of 516 RNA-binding proteins across six cell lines

(differential expression analysis (*t* tests), false-discovery rate corrected with Benjamini-Hochberg procedure) to the amount of potential FL *hTERT* as measured by our splice isoform assays. This analysis revealed 12 genes whose expression significantly correlated to the amount of FL *hTERT* (heatmap shows differential expression of the 12 genes that correlated to *hTERT*

**Fig. 1** *hTERT* alternative splicing is regulated by a network of RNA-binding proteins. **a** *TERT* minigene reporter construct and products. CMV: human cytomegalovirus immediate-early enhancer and promoter, Bp: base pair, B6: variable nucleotide repeat in intron 6 from ref. [13], DR6 and DR8: direct repeat 6 or 8, respectively, from ref. [13], BGH polyA: signal bovine growth hormone polyadenylation signal. **b** *hTERT* steady-state isoform/splicing profile in non-small cell lung cancer cell lines ($n = 3$). **c** Differential expression analysis of splicing factors that correlated with high *hTERT* full-length expression in six non-small cell lung cancer cell lines. Log2 fold change in gene expression between the cell lines. CLK3 and SNRPB appear twice in the heatmap because two different microarray probes were differentially expressed between the high *hTERT* and low *hTERT* lines. **d** Differential expression analysis of splicing factors related to *hTERT* full-length expression, telomerase activity, and *NOVA1* mRNA expression in the seven non-small cell lung cancer cell lines. Log2 fold change in gene expression between the cell lines. **e** Expression of NOVA1 protein and histone H3 protein in normal (HBECs) and cancerous lung cell lines. Representative western blot images are shown ($n = 3$). **f** Expression of *NOVA1* mRNA by RT-ddPCR in a panel of human fetal and adult tissues. RT reverse transcription, ddPCR droplet digital PCR ($n = 3$). **g** siRNA knockdown of *NOVA1* in H1299 and H920 lung cancer lines shifts *hTERT* splice isoform proportions as determined using RT-ddPCR ($n = 6$). **h** siRNA knockdown of *NOVA1* in H1299 and H920 lung cancer lines reduces telomerase activity per cell equivalent (50 cell equivalents, $n = 6$, Student's *t* test set at *$p < 0.05$ for significance). Data are expressed as means and standard error of the mean where applicable. Supplementary data associated with this figure can be found in Supplementary Figure 1

expression in 6 non-small cell lung cancer cell lines; Fig. 1c). Of these 12 genes, 4 genes overlapped with the 93 genes that also changed (twofold increase in reporter minus beta *hTERT* signal compared to control) in the minigene screen (*SNRPB*, *NOVA1*, *U2AF2*, and *CDC40*). *NOVA1* was the top-ranking tissue-specific RNA-binding protein associated with cancer[20,21], and in silico analysis revealed binding sites for *NOVA1* within the *hTERT* locus. Of the 37 proteins that had a greater impact on switching *hTERT* splicing in the minigene assay, NOVA1 was the only factor with an expression profile that was not ubiquitous indicating the potential for therapeutic targeting. Thus, *NOVA1* was chosen for further analysis.

We hypothesized that *NOVA1* may mark a network of splicing factors related to high levels of *hTERT* FL mRNA[22]. Hierarchical clustering analysis (divisive) was used to separate and prioritize cell lines based on telomerase biology (log2-transformed telomerase and percentage of FL *hTERT*) and expression of *NOVA1* mRNA (log2-transformed as measured by droplet digital PCR (ddPCR)). Using two groups of cell lines (seven cell lines total: four with high telomerase, percentage of FL *hTERT*, *NOVA1* compared to three low telomerase, percentage of FL *hTERT*, and *NOVA1*) we compared the expression profiles of the 516 RNA-binding proteins and splicing factors (dendrogam of hierarchical clustering analysis of cell lines Supplementary Figure 1I). This expression analysis revealed 22 genes with different expression patterns between the two groups of lung cancer cell lines (Fig. 1d; heatmap for "Network Discovery Data set"). Next, we probed the overlapping genes between this gene list (22) and the minigene list (93) and observed that 7 genes overlapped (*SNRPD2*, *CWC22*, *NOVA1*, *RBM10*, *ZMAT2*, *SLU7*, and *FAM131B*). Ten genes came out in at least two of the three screens and caused a twofold or greater change in the minigene reporter screen (Supplementary Figure 1J; Supplementary Data file 1 shows tabulated results of all three screens, siRNA sequences, and the overlap analyses).

NOVA1 protein was not expressed in normal human bronchial epithelial cells (HBECs), and was overexpressed in 71% of our lung cancer cell lines (12 of 17 lines; Fig. 1e, Supplementary Figure 1F). We correlated the expression of NOVA1 mRNA and protein to the percentage of *hTERT* FL mRNA and observed a significant correlation in both cases (Pearson's *r*, $R^2 = 0.25$, $p = 0.039$ for mRNA; $R^2 = 0.34$, $p = 0.01$ for protein; Supplementary Figure 1L and M, respectively). *NOVA1* mRNA expression and NOVA1 protein expression were significantly correlated in our non-small cell lung cancer cell lines ($R^2 = 0.50$, $p = 0.001$, Pearson's *r*; Supplementary Figure 1N). We also measured the expression of *NOVA1* mRNA in a panel of human adult and fetal tissues, observing low expression in tissues with the exception of brain (as expected), reproductive (testis), and fetal organs (Fig. 1f).

To confirm the results of the minigene screen and the bioinformatics correlation analysis, we performed short-term siRNA knockdown experiments (Fig. 1g, h). *hTERT* potential FL mRNA levels were reduced by 60% in H1299 and 50% in H920 cells treated with NOVA1 siRNAs compared to cells treated with control non-targeting siRNAs (Fig. 1g). Telomerase enzyme activity was reduced in NOVA1-depleted cells by 2-fold ($p = 0.05$, Student's *t*) and 2.5-fold ($p < 0.05$, Student's *t*) in H1299 and H920 cells, respectively compared to control siRNA-treated cells (Fig. 1h[23]. Knockdown of *NOVA1* was confirmed in both cell lines (Supplementary Figure 1O). Importantly, *hTERT* steady-state transcripts (exons 15/16) were not significantly decreased by transient knockdown of *NOVA1* in either cell line (Supplementary Figure 1P), indicating that NOVA1 knockdown results in a change in splicing and not just a downregulation in transcriptional rate, which is known to affect splicing[24]. We did observe a reduction in exon 7/8 containing transcripts, confirming the reduction in potential FL (Supplementary Figure 1P). These data together with the minigene and bioinformatics analyses support that NOVA1 is a key member of a potential network of genes regulating *hTERT* alternative splicing.

**Long-term depletion of NOVA1 shifts *hTERT* splicing**. Since its initial description, NOVA1 (Ri-antigen) has been observed to be overexpressed in neuro-endocrine cancers such as breast, lung, and brain cancers[20,21,25–27]. NOVA1 contains three K-homology domains and binds RNA at YCAY (Y = C or T) clusters commonly found though out the genome (1 in 64 nts)[28,29]. NOVA1 YCAY RNA-binding motifs are found in *hTERT* introns and exons. We knocked down NOVA1 in three different cell lines: two that express NOVA1 (H1299 and H920) and one that does not express NOVA1 (Calu6, as a control for off-target effects of the short hairpin RNA (shRNA) sequence). Stable knockdown of NOVA1 reduced telomere length and telomerase activity in the two cell lines that express NOVA1 (Fig. 2a, H1299 cells, Supplementary Figure 2A). *NOVA1* protein and mRNA levels were reduced by about 50% (Fig. 2c, Supplementary Figure 2D). NOVA1 knockdown reduced the proportion of FL *hTERT* message (Fig. 2b; Supplementary Figure 2B) and decreased telomerase activity about 50% ($p = 0.05$, Student's *t*, Fig. 2d; Supplementary Figure 2C), which was sufficient to reduce telomere length in both H1299 and H920 cells. The >50% telomerase activity reduction observed leading to progressive telomere shortening is consistent with human diseases of *hTERT* haploinsufficiency that also result in clinically significant shortened telomeres. The long-term depletion of NOVA1 also reduced the steady-state transcript levels of *hTERT* (Fig. 2d, Supplementary Figure 2E). To demonstrate that the shRNA was on target for NOVA1, we transduced H1299 cells with a retroviral 6X MYC-tagged cDNA construct coding for FL *NOVA1* with the shRNA seed sequence

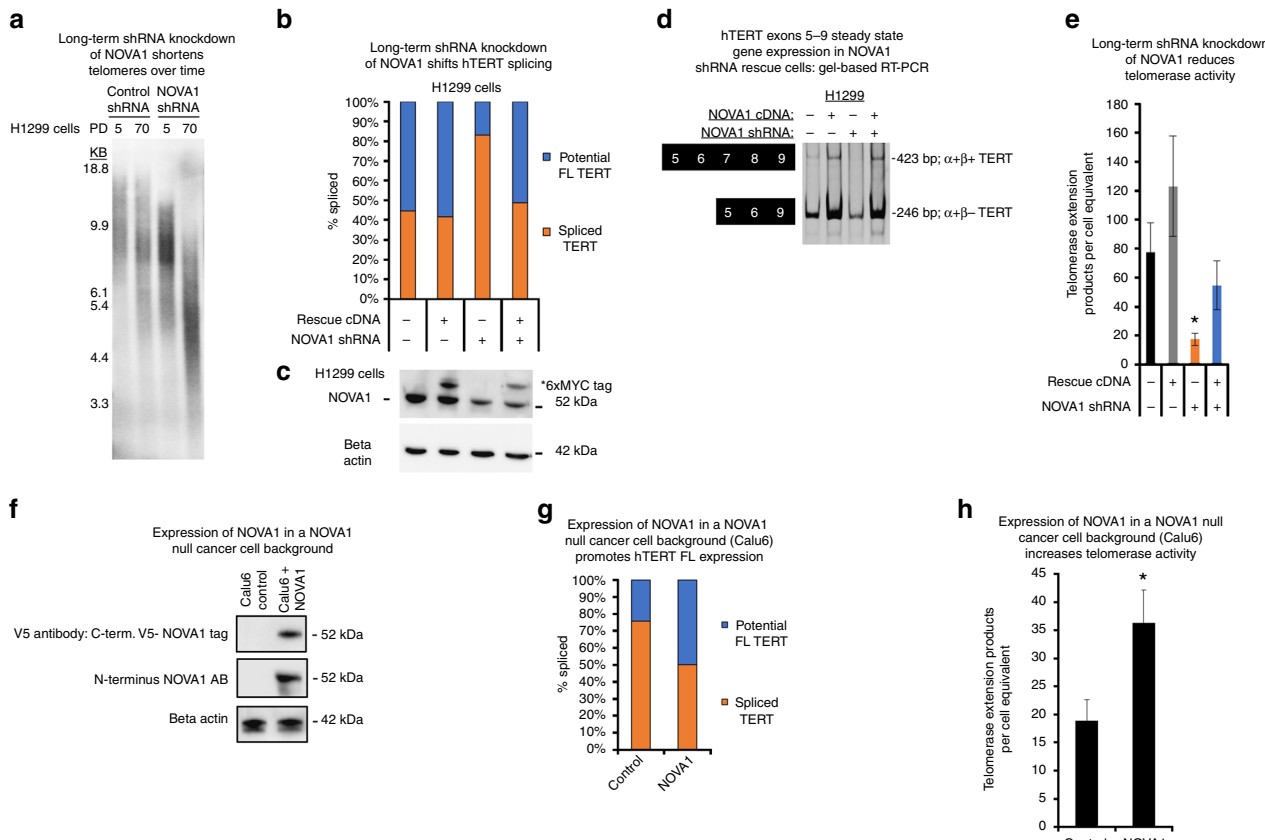

**Fig. 2** Long-term reduction of *NOVA1* progressively shortens telomeres. **a** Terminal restriction fragment length (TRF-Southern blot) analysis of control shRNA or *NOVA1* shRNA at two population doublings (PD). **b** Rescue of shRNA knockdown of *NOVA1* with a shRNA mutant cDNA in H1299 cells (stable cell lines were measured a minimum of six times over several passages). *hTERT* splicing was determined with RT-ddPCR assays. **c** Western blot of *NOVA1* shRNA rescue in H1299 cells (representative image of stable cell lines, measured three times over three passages in culture). **d** *hTERT* expression in rescue H1299 cells as determined by RT-PCR of exons 5–9 (representative image; *n* = 3). **e** Rescue of shRNA knockdown of *NOVA1* with a shRNA mutant cDNA partially restores telomerase activity in H1299 cells (*n* = 6). Telomerase activity was determined with droplet digital TRAP (ddTRAP with 50 cell equivalents added to the assay). Student's *t* test set at *$p < 0.05$ for significance. **f** Western blot of V5-tagged *NOVA1* expression in Calu6 cells (representative image of stable cell lines). **g** *hTERT* splicing profile in Calu6 cells with and without *NOVA1* (*n* = 6). *hTERT* splicing was determined with RT-ddPCR assays. **h** Telomerase enzyme activity (ddTRAP with 50 cell equivalents added to the assay) in Calu6 cells with and without *NOVA1* (*n* = 3). Student's *t* test set at *$p < 0.05$ for significance. Data are expressed as means and standard error of the mean where applicable. *$p < 0.05$. RT: reverse transcription, ddPCR: droplet digital PCR. + indicates presence of shRNA or cDNA construct. − indicates absence of shRNA or cDNA construct (**b**, **d**, **e**). Supplementary data associated with this figure can be found in Supplementary Figure 2

mutated. The shRNA-resistant *NOVA1* cDNA was able to rescue *hTERT* splicing changes, transcript levels, and telomerase enzyme activity (Fig. 2b, d, e). These experiments show that NOVA1 is mechanistically linked to *hTERT* and that the observed phenotypes associated with knockdown are on target. Further, targeting NOVA1 resulted in a potent effect on *hTERT*, not only by shifting the splicing away from FL *hTERT* transcripts but also by reducing overall *hTERT* transcription, which is a different observation compared to the transient siRNAs. The reduced overall *hTERT* mRNA levels are important to consider with recent data indicating that *hTERT* may have non-canonical roles[30,31]. To explain how the NOVA1 rescue returned *hTERT* transcript levels to control values, one possibility is that the increased levels of *hTERT* (Fig. 2d) is due to a transcription factor being spliced or another upstream event that NOVA1 is involved in that results in increased expression of *hTERT*. Recent RNA-sequencing data indicate that NOVA1 may regulate the expression of transcription factors that could be acting upstream of *hTERT* in cancer cells[32].

To determine the effects on cell growth of knocking NOVA1 down in non-transformed lung cells, we introduced control (non-

silencing) and *NOVA1* shRNAs into HBECs and observed no significant growth defect (Supplementary Figure 2M). To further determine the effects of NOVA1 reduction on normal cell growth, we used a siRNA treatment strategy to significantly reduce NOVA1 expression in a normal diploid fibroblast cell strain (BJ fibroblasts). Over the course of the experiment there was no inhibition in growth or viability of the BJ fibroblasts treated with NOVA1 siRNAs compared to control-treated siRNAs (Supplementary Figure 2O and P). The siRNA-induced knockdown of NOVA1 at both the protein (Supplementary Figure 2Q) and mRNA level was significantly reduced compared to control-treated cells (Supplementary Figure 2R).

We attempted to knockout NOVA1 with CRISPR/Cas9 genome-editing methods in H1299 lung cancer cells but were unable to obtain survival clones, suggesting that NOVA1 may be critical in cell survival pathways in H1299 lung cancer cells. When we knocked down NOVA1 with shRNAs in H2882, another non-small cell lung cancer cell line, the cells only divided twice in 90 days following selection (Supplementary Figure 2S). H2882 cells have very robust levels of NOVA1 (Fig. 1e), adding strength to the idea that NOVA1 may be a critical gene to cancer

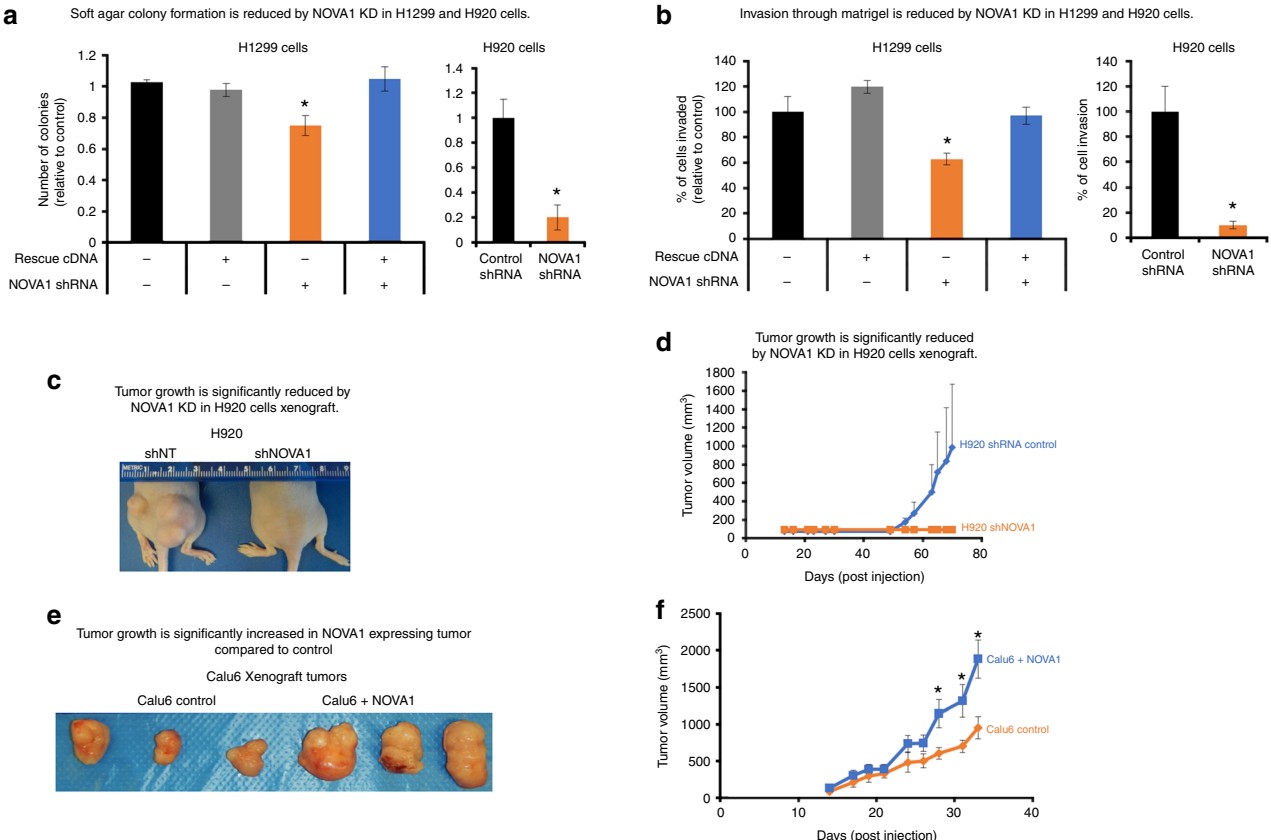

**Fig. 3** Manipulation of *NOVA1* in cancer cells alters xenograft tumor growth. **a** Knockdown of *NOVA1* in H1299 and H920 cells reduced anchorage-independent cell growth compared to controls (soft agar colony formation assays, *n* = 3 for each condition and cell line). + indicates presence of shRNA or cDNA construct. − indicates absence of shRNA or cDNA construct. **b** Knockdown of *NOVA1* in H1299 and H920 cells reduced migration through an extracellular matrix (Boyden Chamber assay) compared to controls (*n* = 3 for each condition and cell line). + indicates presence of shRNA or cDNA construct. − indicates absence of shRNA or cDNA construct (**a**, **b**). **c** Knockdown of *NOVA1* in H920 cells significantly reduced xenograft growth compared to controls. Left panel shows representative images of the hind quarters of nude mice. **d** Growth curves of xenograft tumors during the experiment (*n* = 4 injections for each condition, 2 mice per condition). **e** Expression of *NOVA1* in Calu6 cells results in larger xenograft tumors compared to Calu6 controls (a non-small cell lung cancer cell line lacking *NOVA1* expression). Left panel shows representative images of tumors from Calu6 and Calu6 plus *NOVA1*. Same stable cell line as used in Fig. 2 and Supplementary Figure 2. **f** Right panel shows growth curves of xenograft tumors during the experiment (*n* = 6 injections for each condition, 3 mice per condition). Data are expressed as means and standard error of the mean where applicable. All tests shown are Student's *t* test set at *\*p* < 0.05 for significance. KD: knockdown. Note—an error bar on the controls of **a**–**c** was calculated by first generating the mean of the controls and expressing each control relative to this value. We then calculated a standard deviation and standard error based on the variance observed between these normalized values. Supplementary data associated with this figure can be found in Supplementary Figure 3

cell survival. We observed that a different shRNA targeting NOVA1 in H1299 cells significantly slowed growth and/or was rapidly silenced (Supplementary Figures 2T and U).

**Expression of NOVA1 in cancer cells shifts *hTERT* splicing**. To determine if expression of NOVA1 in a NOVA1-negative cancer cell line promotes *hTERT* FL splicing and increases in telomerase activity, we transduced Calu6 cells with a lentiviral vector containing a V5-C terminus-tagged *NOVA1* FL cDNA. Overexpression of NOVA1 with both tagged (V5 epitope antibody) and N terminus NOVA1 antibodies (Fig. 2f) was confirmed. Telomerase activity increased twofold (*p* = 0.05, Student's *t*, Fig. 2h) as did the proportion of *hTERT* FL transcripts (23% vs. 50%; Fig. 2g) in NOVA1-expressing cells compared to control empty vector cells. In these experiments, telomere length was slightly elongated by overexpression of NOVA1 (Supplementary Figure 2N) in line with the NOVA1 knockdown data.

**NOVA1 knockdown reduces cancer cell growth phenotypes**. We assayed the tumorigenic properties of NOVA1 knockdown

cells. NOVA1 has a known role in breast and lung cancers[33] and is important during normal development for growth, survival, migration, and apoptosis[32,34,35]. NOVA1 knockdown H1299 cells at population 50 formed significantly fewer colonies compared to control and NOVA1 rescue cells (*p* < 0.05, Student's *t*, Fig. 3a). NOVA1 knockdown H1299 cells did not invade as efficiently as H1299 shRNA control cells or the NOVA1 rescue H1299 cells using Boyden chamber assays (*p* < 0.05, Student's *t*, Fig. 3b). Finally, in a colony formation assay we observed that NOVA1-depleted H1299 cells formed fewer colonies compared to control shRNA H1299 cells and rescue cells (*p* < 0.05, Student's *t*, Supplementary Figure 3). We also assayed cancer growth phenotypes in a second cell line after 50 population doublings, H920, and both anchorage-independent growth and invasive phenotypes were significantly reduced in NOVA1-depleted cells compared to control shRNA cells (Fig. 3a, b; *p* < 0.05, Student's *t*). Interestingly, the levels or amount of NOVA1 in the cell lines (H1299 moderate NOVA1 compared to high levels of NOVA1 in H920 cells) seems to correlate the response of the cells to knockdown of NOVA1. For instance, the impact of NOVA1 knockdown on colony formation and migration was much greater in H920 cells compared to H1299 cells. In Calu6

cells, which lack NOVA1 expression, we observed no differences in anchorage-independent growth, invasion, or colony formation between NOVA1 shRNA cells and control shRNA cells, indicating that these effects were due to on-target effects of the shRNA (Supplementary Figure 3).

To directly test if the growth phenotypes observed with NOVA1 knockdown were due to changes in telomere length or telomerase expression, we attempted to rescue the NOVA1 knockdown by expression of an *hTERT* cDNA. We introduced an *hTERT* cDNA (pMIN-Ub-IRES-Blast lentiviral vector from ref. [36]) into H1299 cells that either already had control shRNA or NOVA1 shRNA and were cultured for 50 population doublings. In the H1299 cells with control shRNAs or NOVA1 shRNAs, we confirmed overexpression of *hTERT* at the mRNA level (Supplementary Figures 3D and E) and increased telomerase activity in the *hTERT*-transduced cells (Supplementary Figure 3F). Additionally, we introduced NOVA1 and control shRNAs into U2OS cells, which do not depend on telomerase to maintain telomeres but rather use alternative lengthening (ALT) of telomeres to maintain telomeres. Post selection, these cells were expanded and assayed to assess the impact of NOVA1 knockdown on growth independent of telomere length changes. We confirmed NOVA1 knockdown in the U2OS cells (Supplementary Figure 3J). We assayed each of these model systems (manipulated lines derived from H1299 cells and U2OS cells) for clonogenicity and anchorage-independent growth. We observed that NOVA1 knockdown was not bypassed by forced *hTERT* expression in the H1299 cells with NOVA1 knockdown at population doubling 60 (Supplementary Figure 3H). The H1299 cells with the combination of *hTERT* expression and NOVA1 knockdown formed fewer colonies in both assays of growth compared to H1299 cells with control shRNA and *hTERT* expression (both the NOVA1 knockdown and the NOVA1 knockdown with *hTERT* expression formed about 50% fewer colonies in the colongenicity assay and 40–50% fewer colonies in the anchorage-independent soft agar growth assay; Supplementary Figure 3H and I). Interestingly, upon *hTERT* overexpression the levels of NOVA1 increased in both the shRNA control and shRNA NOVA1 lines. This observation can indicate at least two possibilities; either a hypermorphic phenotype where *TERT* is acting in a way it normally would not due to the high expression levels or that a regulatory loop exists between NOVA1 and *hTERT*. Further, ALT cells with control shRNAs grew substantially better in both growth assays compared to NOVA1 shRNA cells, forming about twice as many colonies in both assays as well (Supplementary Figure 3K and L). This indicates that beyond the control of telomerase activity and *hTERT* splicing in cancer cells, NOVA1 is independently involved in pathways related to tumorigenesis and clonogenic growth.

**NOVA1 confers a growth and survival advantage of tumor cells**. To determine if cell lines with high or low levels of NOVA1 formed tumors in vivo, we injected cells into both hind flanks of immunocompromised mice. We observed three out of four injections of H920 shRNA control cells formed tumors in vivo while only one of four injections of H920 NOVA1 knockdown cells formed tumors (Fig. 3c, d). The tumors derived from the control cell lines were significantly larger compared to the single tumor derived from the NOVA1 knockdown cells. This indicates that *NOVA1* knockdown in H920 cells significantly altered the ability of these lung cancer cells to form tumors in vivo. Further, we injected mice with Calu6 control lentiviral vector cells (a cell line that does not express NOVA1) and compared it to Calu6 cells with ectopic expression of NOVA1. We monitored the tumors for a period of 4 weeks and observed that NOVA1-

expressing cells formed bigger tumors compared to the control Calu6 cells (Fig. 3e, f). Similar to the knockdown experiment, the NOVA1 expression conferred a growth advantage in vivo over cells that lacked NOVA1.

**NOVA1 interacts with *hTERT* pre-mRNA**. NOVA1 binds to pre-mRNAs in a sequence-dependent fashion, binding to YCAY (Y = C or U) motifs in RNAs[37]. First, we looked in silico at the *hTERT* locus for YCAY motifs, focusing on sequences (including introns and exons) from exons 5 to 10 (Fig. 4a). It is not possible to use previous public HITS-CLIP databases from mice because of sequence element differences between mice and humans[13]. Since NOVA1 is known to bind to clusters of YCAY motifs we looked for areas of highly concentrated motifs. We found several potential candidate regions in *hTERT* exons and introns 5 through 10 (Fig. 4a). We performed ultraviolet (UV)-crosslinking and immunoprecipitation (UV-IP)[38] of H1299 cells and observed an enrichment of the NOVA1:*hTERT* RNA interaction in *hTERT* intron 8 (direct repeat 8, DR8). NOVA1 appeared to bind in a region we previously observed to be involved in the regulation of *hTERT* alternative splicing (Fig. 4a–c)[13]. Our working model is that NOVA1 binds to the DR8 region and promotes splicing of *hTERT* to include RT domain-coding exons 7 and 8. This idea is consistent with previous observations suggesting that NOVA1 can act as a splicing enhancer if it binds following an alternatively spliced exon[39]. From in silico analysis we found that DR8 of *hTERT*, a 258-base pair element, contains 7 YCAY NOVA1-binding motifs. Previously, we found that when cells were treated with a 2-O-methyl-antisense oligonucleotide to this region, *hTERT* minus beta splicing was increased and FL splicing was reduced, supporting the idea that a factor that promotes inclusion of *hTERT* exons 7 and 8 was blocked[13]. This is consistent with our observations that *hTERT* minus beta splicing increases when NOVA1 levels are reduced. To confirm the UV-IP observations, we utilized our *hTERT* minigene series that we previously developed, which excludes DR8, the intron 8 region containing the NOVA1-binding motifs, and performed crosslinking and immunoprecipitation followed by RT-ddPCR (CLIP-ddPCR) (Fig. 4d). As additional controls, we included constructs that contained or excluded a highly similar (85% homology) sequence region in intron 6 called direct repeat 6 (DR6; Fig. 4e). Interestingly, DR6, a 254-base pair region in intron 6, contains fewer YCAY (5) motifs than DR8. When DR8 was present we could effectively pull down *hTERT* minigene pre-mRNAs, but we could not when DR8 was absent. As a control, we also assayed our CLIP cDNAs for a known NOVA1 target gene, glycine receptor alpha 2 (*GLRA2*)[40]. As previously described *GLRA2* exon 3A and 3B are mutually exclusive exons regulated by NOVA1. First, we tested for expression of *GLRA2* mutually exclusive exon 3A and 3B usage in HeLa and H1299 cancer cells (Supplementary Figure 4C and E), and observed that exon 3B of *GLRA2* was preferentially used in cancer cells. Next, we looked in our H1299 *NOVA1* rescue series to see if *GLRA2* was regulated by NOVA1 in cancer cells and indeed found that GLRA2 exon 3B is preferentially used when NOVA1 levels are higher (Supplementary Figure 4D) similar to previous studies[40]. Next, we assayed our CLIP cDNAs and observed that *GLRA2* was effectively pulled down in all extracts regardless of *hTERT* status, indicating that our CLIP was efficient (Supplementary Figure 4E). To further confirm the CLIP and UV-IP observations, we in vitro transcribed a 1 kb fragment of RNA containing DR8 of *hTERT* intron 8 and performed an RNA pull-down assay (Fig. 4f). When we expressed NOVA1 in 293 cells and exposed the lysate to the labeled RNA, we observed effective pull down of NOVA1 (Fig. 4g). These data indicated that NOVA1 was binding to this 1 kb RNA fragment of *hTERT* intron

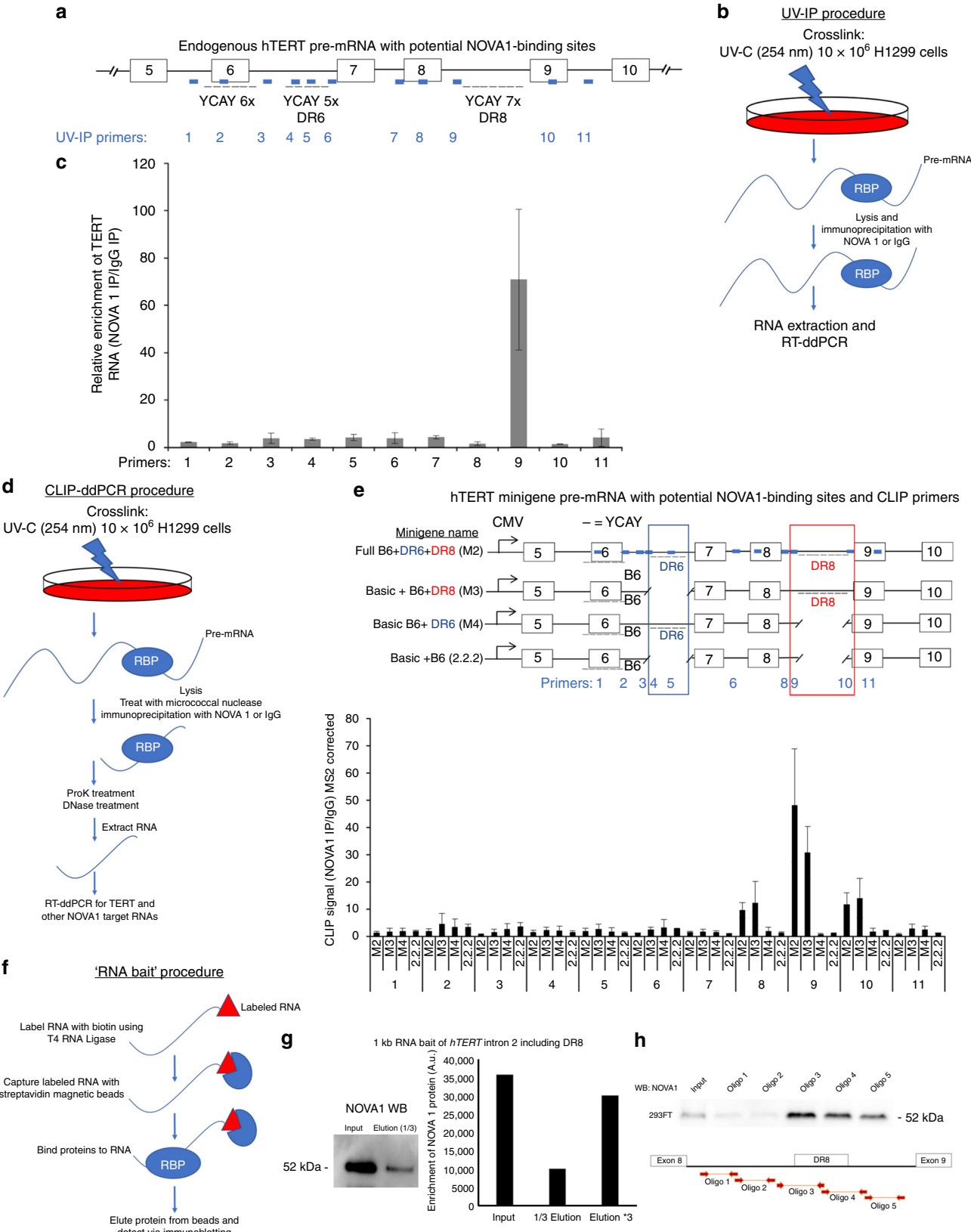

8. Since a 1 kb RNA fragment may bind many proteins, we generated a series of smaller (~150 nt) RNA baits surrounding and in DR8 of *hTERT* to more specifically determine where NOVA1 was binding. We observed binding to oligos 3, 4, and 5 (Fig. 4h) and weak to no binding to oligos 1 and 2 (Fig. 4h), with the strongest binding to oligo 3 in the 5′ end of DR8. We hypothesize that NOVA1 is binding to DR8 and the region surrounding DR8 of *hTERT* through both direct and indirect interactions (that is, as part of a splicing factor complex) that directs NOVA1's impact on *hTERT* splicing. There are 7 YCAY

**Fig. 4** *NOVA1* binds to a deep intronic element in *TERT* precursor RNAs. **a** Cartoon of *TERT* exons 5 through 10 showing potential *NOVA1*-binding sites and the primers (blue boxes) used in the ultraviolet immunoprecipitation (UV-IP) procedure. **b** UV-IP procedure schematic showing the major steps. **c** Droplet digital RT-PCR showing UV-IP enrichment of *NOVA1* at *TERT* DR8 ($n = 3$ independent IPs). **d** Schematic of UV-crosslinking immunoprecipitation (CLIP) droplet digital PCR procedure. **e** Cartoon of *hTERT* minigenes used in the CLIP experiments, primers (light blue boxes), and droplet digital PCR quantification of *NOVA1* protein and *TERT* RNA interaction ($n = 3$ independent IPs). Differences in constructs are highlighted in red and dark blue boxes and text. The minigenes where plasmid DNA was removed to generate the different constructs lacking the direct repeat regions are shown by a broken line. **f** Schematic of RNA bait procedure to find proteins that interact with *hTERT* RNAs. A second set of RNA baits were made from PCR fragments of *hTERT* intron 8 around and including DR8. **g** We in vitro transcribed a 1 kb fragment of *hTERT* intron 8 that contained DR8. Western blot and quantification of western showing pull down of *NOVA1* protein with *TERT* RNA containing DR8 ($n = 2$). **h** Representative western blot of *NOVA1* protein showing binding to DR8 at baits (oligos) 3, 4, and 5 ($n = 2$). Data are expressed as means and standard error of the mean where applicable. *$p < 0.05$. Supplementary data associated with this figure can be found in Supplementary Figure 4

motifs in DR8 (oligos 3 and 4), oligo 5, which is 71 nts 3′ of DR8 contains an additional 2 YCAY motifs. We hypothesize that a complex of proteins, including NOVA1, that interact at or near DR8 could be pulling down NOVA1 with oligo 5, explaining why NOVA1 is present in oligo 5's pull down. Overall, our results are consistent with the model that when NOVA1 is bound to *hTERT* mRNAs in intron 8, it acts as a splicing enhancer, promoting the inclusion of exons 7 and 8 to increase the production of FL *hTERT* mRNAs.

**Deletion or mutation of NOVA1 binding shifts *hTERT* splicing**. To further define the importance of *hTERT* DR8 in telomere biology, we deleted a 480-nt fragment of *hTERT* intron 8 using two CRISPR guide RNAs flanking DR8 (Fig. 5a). We identified three H1299 clones that had the correct on-target deletion of *hTERT* intron 8 DNA containing DR8 (Supplementary Figure 5A, B and C). In a parallel experiment, we introduced either wild-type (WT) or mutant DR8 DNA via a donor plasmid with CRISPR/Cas9 (Fig. 5b). The mutant donor plasmid had all seven of the YCAY motifs in DR8 mutated to YAAY, which has previously been shown to block NOVA1 recognition of target genes[34]. We identified 5 clones with homozygous integration of the mutant NOVA1-binding sites and sorted an additional 6 single-cell WT clones to control for heterogeneity of tumor cell lines[41] (Supplementary Figure 5E). This produced 15 cell lines (14 clones and the parental H1299 population) that we then followed over time for telomere length, telomerase activity, and *TERT* splicing phenotypes.

We observed clonal heterogeneity for telomere length between all the clones, including the control clones, as expected. In the sorted control clones, however, we observed minimal telomere length changes over time ($14 \pm 4.4$ nts per doubling; average ± standard deviation), while significant telomere shortening was observed in both DR8 YAAY mutant clones ($46 \pm 42$ nts per doubling; $p = 0.05$) and in the DR8-deleted clones ($108 \pm 45$ nts per doubling; $p < 0.001$) (Fig. 5c). All the DR8-deleted clones had longer telomeres on average compared to the mutant DR8 clones. Thus, deletion of DR8 may produce a strong selection pressure for clones with longer telomeres. *hTERT* splicing was also significantly changed, with a dramatic shift toward spliced products (35% FL in sorted controls versus 12% FL in DR8 YAAY mutants, $p < 0.001$, Student's $t$; Fig. 5d, Supplementary Figure 5G). Further, we assayed telomerase enzyme activity over time at three different population doublings and observed that on average, the DR8 YAAY mutants had 70% less telomerase activity compared to the controls while the DR8 deletion clones had nearly undetectable telomerase activity (Fig. 5f, Supplementary Figure 5G). The telomere shortening rates of 46 nts per doubling and 107 nts per cell doubling in the DR8 mutants (70% telomerase inhibition) and DR8-deleted clones (nearly 100% telomerase inhibition) correlate closely with observations that

telomerase adds 50–150 nts per cell division to maintain telomeres[42].

We also observed clonal heterogeneity for *hTERT* splicing as shown in Fig. 5e: three clones had mostly minus beta, one clone had higher levels of *hTERT* FL and total *hTERT* mRNA, which correlated to higher telomerase activity (DR8 mutant 5), and one clone seemed to lack all transcripts with exons 5–9 (DR8 mutant 3, Supplementary Figure 5G and H). The clone (DR8 mutant 3) that completely lacked mRNA containing *hTERT* exons 5–9 also had no detectable telomerase activity (via TRAP) and eventually died in culture at population doubling 58 post sorting. Further, only one of three DR8-deleted clones showed FL *hTERT* message and telomerase activity (DR8 deletion 2, Supplementary Figure 5G and H). After long-term passage one of the DR8 deletion clones stopped dividing at PD 85 (DR8 deletion 1). We measured the expression level of *NOVA1* mRNA in these clones and observed that on average the clones were similar for *NOVA1* mRNA expression levels (Supplementary Figure 5I), however we observed heterogeneity in expression between clones with different telomere biology phenotypes. For example, the deletion clone that died (DR8 deletion 1) and the mutant clone that died had low expression of *NOVA1* compared to the average (Supplementary Figure 5). In contrast, the DR8 deletion clones with telomerase activity (DR8 deletion 2) had very high *NOVA1* levels. This suggests that *NOVA1* may help to maintain telomerase levels via an alternative binding site that promotes *hTERT* FL splicing. Overall, these data solidify the role of DR8 in the splicing choice of *hTERT* to produce FL or spliced products.

To determine if DR8 was important only in cell lines with NOVA1 or as a general splicing enhancer region, we compared the response of H1299 cells (cells that express NOVA1) and Calu6 cells (cells lacking NOVA1 expression) following treatment of an antisense oligonucleotide that blocks a sequence 19 nts from the 5′ end of DR8 (including a YCAY element, antisense oligo called, "DR8 + 19") and was previously shown to shift *hTERT* splicing toward minus beta containing transcripts in a Hela cell clone that expresses NOVA1[13]. H1299 cells treated with 50 nM of DR8 + 19 antisense oligo for 48 h significantly shifted *hTERT* splicing toward minus beta while Calu6 cells treated did not shift the splicing of *hTERT* (Supplementary Figure 5J and K). This indicates that DR8 is likely only an important splicing enhancer region in *hTERT* when NOVA1 is present.

**Discussion**

The catalytic protein component of telomerase, *hTERT*, is spliced into multiple isoforms, but only FL *hTERT* mRNA is capable of producing enzymatically active protein that can maintain telomeres[13]. We observed that ~70% of the non-small cell lung cancers tested here express moderate to high levels of NOVA1. NOVA1 expression promotes the inclusion of exons 7 and 8 in the RT domain of *hTERT* to produce enzymatically active telomerase and regulate telomere length (Fig. 6). Further, NOVA1

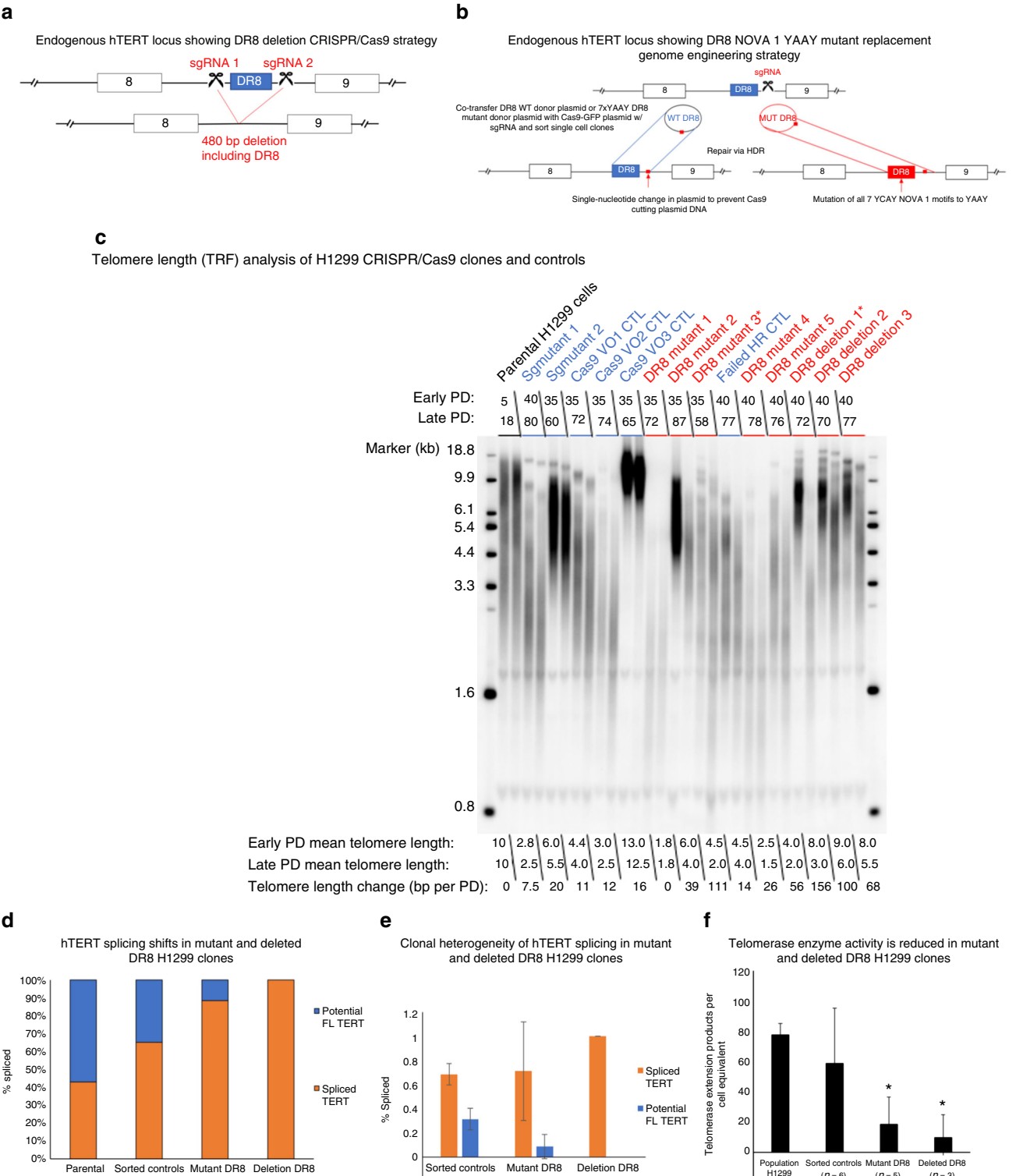

**Fig. 5** Deletion of DR8 or mutation of *NOVA1*-binding sites in *TERT* shortens telomeres. **a** Schematic of CRISPR/Cas9 strategy to delete DR8 in *TERT* intron 8. **b** Schematic of CRISPR/Cas9 strategy to mutate *NOVA1*-binding sites in *TERT* DR8. **c** Terminal restriction fragment length (TRF-Southern blot) analysis of parental, sorted controls, DR8 *NOVA1* '7 × YAAY' mutants, and DR8-deleted H1299 clones. *DR8 mutant 3 and *DR8 deletion 1 stopped growing at population doubling 57 and 85, respectively. **d** *hTERT* splicing isoform proportion analysis of parental, averaged sorted controls ($n = 6$), DR8 mutants ($n = 5$), and averaged DR8 mutants ($n = 3$). **e** *hTERT* splicing isoform proportion analysis of averaged sorted controls ($n = 6$), averaged DR8 mutants ($n = 5$), and averaged DR8 mutants ($n = 3$). This plot shows the clonal heterogeneity of *hTERT* splicing in the clones used in this experiment. Standard deviation of the group means displays the variation in the response to mutation or deletion of direct repeat 8 (DR8) in *hTERT*. Only sorted controls and clones are shown as variation in the parental line is shown elsewhere. **f** Telomerase activity of parental, averaged controls ($n = 6$), averaged DR8 mutants ($n = 5$), and averaged DR8 deletion clones ($n = 3$). Data are expressed as means and standard deviation where applicable. *$p < 0.05$. Supplementary data associated with this figure can be found in Supplementary Figure 5

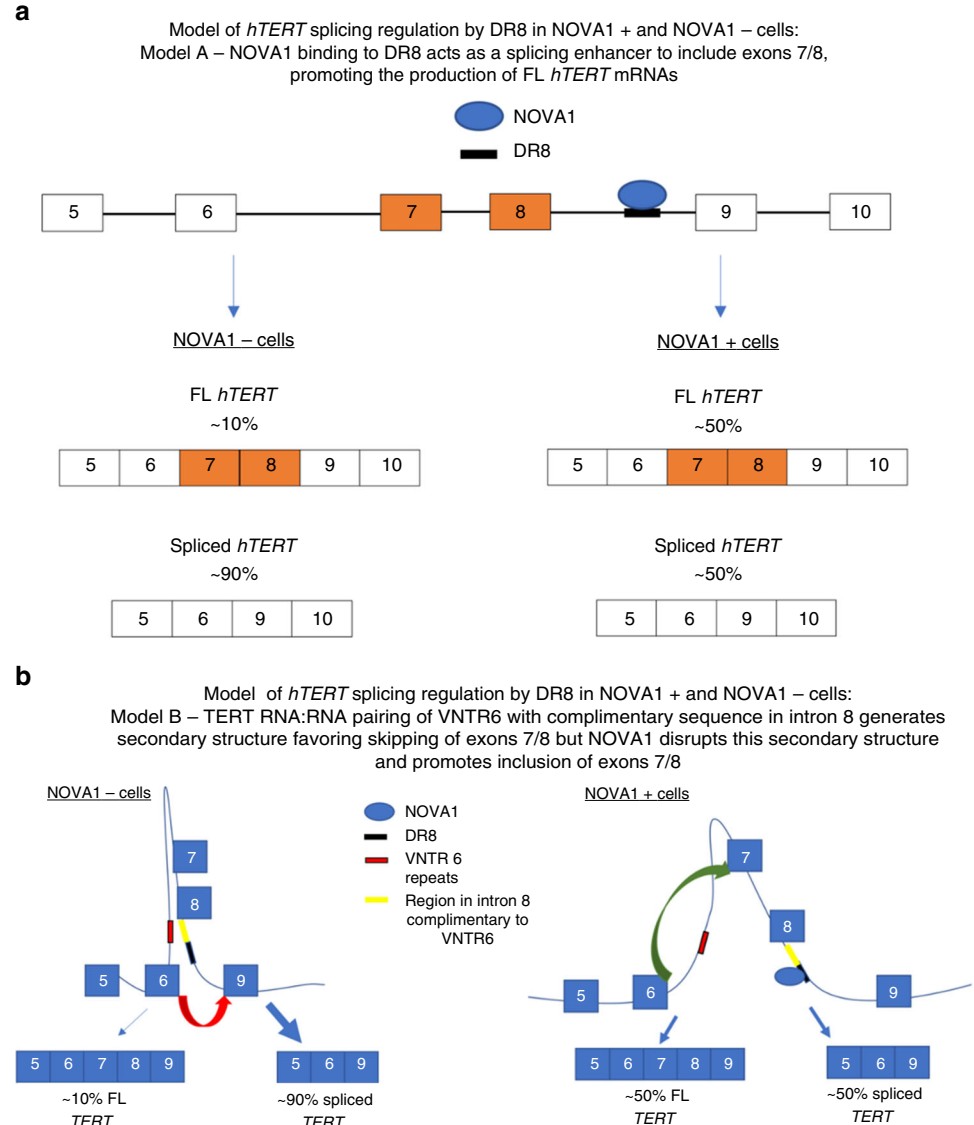

**Fig. 6** *NOVA1* binds to a deep intronic sequence in *hTERT* DR8 and acts as a splicing enhancer to promote the inclusion of exons 7/8 and greater levels of full-length *hTERT* mRNAs. **a** Linear model depicting *NOVA1* binding to *TERT* DR8 promoting the inclusion of exons 7/8. **b** Alternative model of *NOVA1* promoting full-length *TERT* considering RNA:RNA pairing/RNA secondary structure of *TERT* pre-mRNAs

may be a regulator of growth and invasion-related signaling in cancer cells, coupling telomere length maintenance to other cancer cell characteristics (Fig. 3). Thus, the applicability of the *hTERT* minigene screen data we have compiled represents a valuable resource that could be mined to identify other splicing factors that could be targeted in cancer.

These studies extend earlier findings that RNA metabolism is a regulatory component of telomerase activity and telomere length maintenance[43,44]. For instance, the mRNA decay pathway was found to be critical in telomere length maintenance in yeast[45]. Additionally, our splicing factor screen and the work of others point to the importance of hnRNP proteins in telomere biology and in the regulation of *hTERT* pre-mRNA splicing[46–51] (Fig. 1). We confirmed factors that were previously reported to directly bind to *hTERT* pre-mRNA and regulate minus beta splice choice (SRSF11, hnRNPH2, and hnRNPL)[15]. We observed significant changes in the minigene reporter assay for hnRNPH1, hnRNPF, and hnRNPM, which were all observed to potentially bind *hTERT* pre-mRNAs in CLIP experiments[52], further validating the minigene data. As far as we could determine, this is the first study to

investigate knockdown of splicing factors in human cancer cells with regard to telomere biology and we observed that even modest knockdown is compatible with long-term tissue culture studies to investigate changes in the telomere length of cells.

During the generation of CRISPR/Cas9 clones of H1299 lung cancer cells some striking and unexpected phenotypes arose upon mutation and deletion of DR8 in *hTERT*. Two clones that eventually died off in culture had about 2 kb telomere lengths when they died (DR8 mutant clone 3 and DR8 deletion clone 1). However, DR8 mutant clone 4 had a mean telomere length of about 2 kb and survived in culture (Fig. 5). We speculate that the telomere DNA damage sensing capabilities between the clones that died compared to the clones that survived at these short telomere lengths are different. Further, DR8 mutant clone 3 and DR8 deletion clone 1 both lacked expression of *hTERT* exons 5–9, while DR8 mutant clone 4 had a small amount of FL *hTERT* remaining (Fig. 5). This small amount of *hTERT* FL could have generated enough active telomerase to maintain the short telomeres. We speculate DR8 mutant clone 4 could have had recruitment and processivity advantages compared to other lines

with higher telomerase activity but increased rates of telomere shortening. These clones could be useful in future studies aiming at elucidating the genetic and molecular underpinnings of these less-well-described phenomena.

We propose two models for how the *NOVA1* protein is interacting with *TERT* pre-mRNAs (Fig. 6). In the first model only considering NOVA1 protein's influence on *hTERT* pre-mRNAs, we propose that NOVA1 binds to DR8, recruiting the basal exon junction recognition machinery that promotes the use of exon 8's 5′ splice donor with exon 9's 3′ splice acceptor site to generate *hTERT* mRNAs, including exons 7 and 8 (Fig. 6a). In the second model, we consider both pre-mRNA secondary structures and NOVA1 proteins. We previously defined a role for RNA:RNA pairing in regulating minus beta splice choice[14]. In this model (Fig. 6b), RNA:RNA pairing of a variable nucleotide tandem repeat in intron 6 of *hTERT* creates an RNA secondary structure that brings the exon 6 5′ splice site in close proximity to the exon 9 3′ splice site, generating a structural change where joining of exon 6 to exon 9 is favored over exon 6 to exon 7, producing the minus beta splice variant over FL mRNAs. When NOVA1 is present and bound to DR8, this disrupts the secondary structure created by RNA:RNA pairing between intron 6 and intron 8 and brings the exon 8 5′ splice site closer to the exon 9 3′ splice site thus producing more FL molecules of *hTERT* compared to cells lacking NOVA1 (Fig. 6b). Supporting this model, when DR8 is deleted or all seven YCAY motifs in DR8 are mutated three lines switched their splicing to almost solely minus beta (Supplementary Figure 5). In one DR8 mutant (clone 3) and one DR8 deletion (clone 1) we observed a complete loss of expression of *hTERT* exons 5–9 (Supplementary Figure 5). One possibility is that the loss of NOVA1 binding in these clones leads to a change in secondary structure in the pre-mRNA of *hTERT* and results in a complete skipping of exons 5–9 but that other regions of *hTERT* are still expressed (Supplementary Figure 5H).

The present studies add to the growing body of literature that alternative splicing may be targetable in cancer[53]. The present study points out that telomerase activity and telomere length maintenance can be manipulated by targeting the alternative splicing machinery. We chose to follow-up on NOVA1, a well-defined splicing factor that has implications in cancer biology as well, because it had binding sites in *hTERT* introns and exons, and because it displays cancer cell-specific expression across a wide variety of human tissues (Fig. 1, besides neurons and testis). In addition to NOVA1's role in neurons, NOVA1 can also have a role in the survival of cancer cells (Fig. 3). It is unlikely that NOVA2 is playing a similar role in cancer biology as it is only expressed in brain cancers and absent in nearly all other human tumors (http://www.proteinatlas.org/ENSG00000104967-NOVA2/cancer).

It is clear that NOVA1 has additional targets in cancer cells that are related to the growth phenotypes observed herein. NOVA1 may regulate AKT levels influencing growth and survival of cancer cells. The role of AKT in cell proliferation and survival is well documented and several anticancer drugs are currently in use that target AKT. Further, AKT is upstream of FOXO transcription factors, which controls cell survival and apoptosis. AKT functions in cellular survival by phosphorylating and sequestering FOXO's in the cytoplasm and keeping them inactive and unable to initiate transcription of the cell death program. When NOVA1 levels are reduced, AKT protein levels are reduced[32], FOXO's become active and initiate transcription of the cellular death machinery. Future studies investigating the role of NOVA1 in AKT induced cancer cell survival are warranted. Further, AKT is also a kinase that may increase the nuclear pool of TERT proteins and thus increase the potential for active telomere length maintaining telomerase[54]. Thus, a NOVA1-AKT-TERT axis may exist in some cancers that may be targetable.

Overall, our results are consistent with the idea that *hTERT* splicing is malleable in cancer cells by either targeting the intronic DR8 region or by targeting NOVA1. While NOVA1 impacts *hTERT* pre-mRNAs, it also has other targets in cancer cells. Further studies will be required to determine if NOVA1 is indeed clinically targetable in cancer, including the identification of proteins that interact with NOVA1 and the genes that NOVA1 regulates in cancer. In summary, NOVA1 appears to integrate replicative immortality (telomerase activity and telomere length maintenance), with sustained proliferative signaling, invasion, and metastasis. Thus, the present studies have provided new insights into *hTERT* regulation and identified *NOVA1* as a lead candidate gene for development of small-molecule inhibitors targeting the splicing machinery for anticancer therapy.

## Methods

**Plasmids**. Human telomerase (*hTERT*) exon and intron sequences were inserted into pcDNA5/FRT expression vector (Invitrogen, Carlbad, CA) and were modified to exclude exons 5 and 10 and introns 5 and 9 to shorten the construct[13]. The *hTERT* minigene was modified with renilla and firefly luciferase fragments. The minigene was integrated into a single locus using previously described methods[13,14]. We modified the previously used minigene to include a renilla (*Rluc*) and a firefly luciferase (*Fluc*) gene. We fused renilla luciferase and E2A (*equine rhinitis* A virus 2A) peptide in frame to *hTERT* exon 8 so that when exon 8 was included in the minigene pre-mRNA, the protein product would produce *Rluc* (Fig. 1a) putting the *Fluc* out of frame. Conversely, when exons 7 and 8 are skipped and splicing occurs from exon 6 to exon 9, *Fluc* and a T2A (*Thosea asigna* virus 2A) peptide are in frame. The use of the 2A peptides facilitated the generation of equal molar ratios of luciferase to *hTERT*[55]. Lentiviral plasmid for secreted embryonic alkaline phosphatase (SEAP, Clontech) Lentiviral GFP pGIPZ shRNA plasmids for control (non-targeting), *NOVA1* (Openbiosystems, NOVA1–5′-TTGGACTTAGACAG CTTGA) were obtained. Lentivirus was made by co-transfecting 5 μg of proviral shRNA plasmids and 2 μg of packaging plasmids pMD2.G and psPAX2 using Polyjet transfection reagent (SignaGen Laboratories) into 293FT cells. CCSB-Broad lentiviral human *NOVA1* FL cDNA with a C-terminal V5 tag and blasticidin selection in mammalian cells (accession: BC075038, clone ID: ccsbBroad304_01104) was purchased and sequence verified by our group (GE, Dharmacon). Viral particles were produced as above. ShRNA-resistant *NOVA1* cDNA was generated by site-directed mutagenesis (Agilent Quickchange). Retroviral particles were generated in 293FT packaging cells with pUMVC and pVSVG. Plasmids for CRISPR/Cas9 experiments are described below in the genome engineering section.

**Cell culture and cell lines**. All non-small cell lung cancer cell lines (H1299, H920, Calu6, and H2882) were cultured at 37 °C in 5% $CO_2$ in 4:1 DMEM:Medium 199 containing 10% calf serum (HyClone, Logan, UT). Briefly, HBECs were maintained in low oxygen conditions in serum-free media containing supplements from the Keratinocyte-SFM media (Invitrogen/Gibco catalog # 17005-42) on a collagen/gelatin-coated tissue culture dish[56]. All cell lines were obtained from American Type Culture Collection, or as a kind gift from Drs. John Minna and Adi Gazdar.

**Cancer growth assays**. Stable knockdown cells and controls (non-targeting shRNA) were suspended in 0.375% Noble agar (Difcon, Detroit) in supplemented basal medium at two densities (1000 and 2000 cells) and overlaid on 0.75% Noble agar in 24-well plates. Each density was seeded in triplicate and each assay was performed twice. Colony formation efficiency was calculated by the average number of colonies counted per cell divided by the number of cells seeded. Colonies larger than 0.1 mm were measured and counted after 10 days of growth and the average of the counts was used. Data are plotted as fold change over non-targeting shRNA cells. Data were analyzed with two-tailed Student's *t* tests[57].

Invasion was determined using Boyden chamber assays. Briefly, cells were serum starved overnight (~16 h) prior to assays for invasion. Twenty-four-well Matrigel-coated transwell filters (BD Biosciences, San Jose, CA) were thawed and rehydrated according to the manufacturer's instructions. Cells were collected and re-suspended in serum-free media and added to the top chamber in duplicate. The bottom chamber was filled with 2% serum-containing media (4:1 DMEM:Medium 199) as a chemoattractant. Cells were incubated overnight. Non-invaded cells were scraped off with a cotton swab and wells were washed with phosphate-buffered saline (PBS). Invaded cells were fixed for 5 min with 10% neutral buffered formalin and stained for 10 min with Hoechst (Invitrogen). Images were taken at ×10 magnification[58].

For colony formation assays, cells were plated at clonal density (30–70 cells per 2.5 cm² dish) in 10% serum-containing media. Cells were analyzed 7 days after plating by staining with Hoechst (Invitrogen)[59].

To test the dependence of observed growth defects on telomere length and telomerase activity, an *hTERT* cDNA (pMIN-Ub-IRES-Blast lentiviral vector from

ref. [36]) was transdued into H1299 cells with control shRNAs or NOVA1 shRNA after 50 population doublings. This vector contains a 3.398 kb cDNA of *hTERT* coding sequence thus no splicing is needed to make hTERT mRNA. Further information about the *hTERT* cDNA from pGRN145 that was inserted into a lentiviral vector can be found here (https://www.atcc.org/Products/All/MBA-141. aspx#generalinformation). It is clear that *TERT* RT domain was expressed in our experiments and that telomerase was active (see Supplementary Figure 5). Following infection and selection, cells were plated as described above in the cancer growth assays.

**Xenograft.** All animal experiments were approved by the University of Texas Southwestern (UTSW) Institutional Animal Care and Use Committee (IACUC) and conducted as per the institutional guidelines. Athymic NRC nu/nu nude mice (~4–6 weeks old, Charles River) were purchased. Tissue culture cells from H920 control shRNA, H920 *NOVA1* shRNA, Calu6 control (WT with empty lentiviral vector), and Calu6 plus *NOVA1* cDNA were cultured and injected subcutaneously into the hind flanks. For H920 cells 5 million cells in 100 μL of 1× PBS were injected. For Calu6 cell 1 million cells in 100 μL of 1× PBS were injected. Tumor growth was monitored by caliper measurement once or twice weekly. Tumor volume was calculated (volume = (width)$^2$ × length/2).

**Transient siRNA experiments.** For transient knockdown experiments cells were plated in six-well plates (150 000 cells per well) and were transfected with non-silencing controls (Santa Cruz Biotechnology, sc-37007) or a pool of three siRNAs targeting *NOVA1* (Santa Cruz Biotechnology, sc-42142: sense RNA sequences—(1) 5′-GACAGACAAUUGUUCAGUUtt-3′, (2) 5′-GAACGGUUGAAGCACU-GAAtt-3′, (3) 5′-GACCACCGUUAAUCCAGAUtt-3′). Cells were plated 24 h prior to transfections. On the day of transfection, media was switched to 2% serum and transfection complexes were prepared with 50 nM of siRNAs using MEM (Gibco, Invitrogen) and RNAi max (Invitrogen) following the manufacturer's procedures. Following 72 h of exposure to siRNAs cells were washed, trypsinized, counted, and pelleted for RNA extraction and telomerase activity assays. We treated normal BJ fibroblasts three times (every 96 h) with non-targeting control or NOVA1 targeting siRNAs over the course of 12 days, counted the cells, and determined viability with trypan blue staining.

**Western blot analysis.** Total protein lysates were extracted from tissue culture cells using Laemmli buffer, boiled, and the protein concentration determined (BCA protein assay, Pierce). Thirty micrograms of protein was resolved on SDS-polyacrylamide gel electrophoresis gels, transferred to polyvinylidene fluoride membranes, and detected with a rabbit monoclonal antibody for *NOVA1* (Abcam, EPR13847, ab183024, 1:1000 dilution in 5%BSA)). Protein loading was determined with antibodies against with beta actin (Sigma; 1: 20 000 dilution in 5% non-fat dry milk) or histone H3 (Sigma; 1:1000 in 5% non-fat dry milk). Uncropped images of all western blots can be found in Supplementary Figure 7.

**RT-droplet digital PCR.** Tissue panel RNAs were purchased (Clontech, 20 tissue panel II). Three sets of cDNAs were made with a 1:1 mixture of random hexamer and oligo-dT priming with three different RTs: (1) iScript advanced (42 °C, Bio-Rad); (2) Superscript III (55 °C, Invitrogen); and (3) AMV (50 °C, Invitrogen). All RNA samples were spiked with a known amount of MS2 bacteriophage RNA to enable normalization of absolute molecule counts from ddPCR. For tissue panel *hTERT* and *NOVA1* mRNA analysis we used three RTs because we observed differences in detection of *hTERT* using different RTs so to be able to eliminate spurious measures of low abundance targets we averaged data for all three RTs. All cDNAs were diluted 1:4 before use and stored at −80 °C. For *hTERT* splicing analyses we used iScript Advanced (Bio-Rad) to generate cDNAs, diluted 1:4, and used within 48 h of production in ddPCR measures. Primer sequences for *TERT* are listed in Supplementary Table 2. Uncropped gel images of RT-PCR gels can be found in Supplementary Figure 7.

**Droplet digital TRAP assay (telomerase activity).** Quantitation of telomerase enzyme activity was performed using a modified telomeric repeat amplification protocol[23]. Briefly, cells were lysed, diluted, and added to the telomerase extension reaction for 40 min followed by heat inactivation of telomerase. An aliquot of the extension products was amplified in a ddPCR for 40 cycles and fluorescence measured and droplets read and counted on the droplet reader (QX200, Bio-Rad). Following, data were processed and telomerase extension products per cell equivalents determined.

**Telomere length analysis.** The average length of telomeres (terminal restriction fragment lengths) was measured as described in ref. [60] with the following mod-ifications. DNA was transferred to Hybond-N+ membranes (GE Healthcare, Piscataway, NJ) using vacuum transfer. The membrane was briefly air-dried and DNA was fixed by UV-crosslinking. Membranes were then probed for telomeres using a digoxigenin (DIG)-labeled telomere probe, detected with an horseradish peroxidase-linked anti-DIG antibody (Roche), and exposed with CDP-star

(Roche)[61]. Uncropped scans of Southern blot images can be found in Supple-mentary Figure 7.

**Minigene screen setup and reporter assays.** We used two databases (NCBI gene and Genecards) and searched the key words of "RNA binding protein" and "splicing factor" to generate a list of 516 genes (Supplementary Data file 1) and then ordered pools of four siRNAs to each gene (Dharmacon, GE, sequences of pools to each gene are located in Supplementary Data file 1, we had 528 pools of siRNA as several genes had multiple siRNA pools). HeLa cells harboring the *hTERT* minigene splicing reporter were plated and transfected 24 h later with 1 nM of each individual pool of siRNAs. A low concentration of siRNAs (1 nM) helped to reduce the potential viability effects and "off-target" effects. Plates were trans-fected using RNAi max (Invitrogen; sequences in Supplementary Data file 1) and cells were analyzed 72 h following transfection. As negative controls, we used a pool of scrambled siRNAs (siRNA control), a transfection control (cells, transfection reagents, and media), and a "cells only" control (cells plus culture media). Knockdown of the core splicing factor *hnRNPH1* resulted in a fourfold induction of minus beta splicing in the minigene luciferase assay ($p < 0.05$, Student's t; Sup-plementary Figure 1A). *hnRNPH1* was included as a minus beta inducing positive control and ubiquitin (*UBB*) as a transfection-positive control. Cells were lysed in passive lysis buffer and analyzed for renilla and firefly luciferase following the manufacturer's instructions after 72 h (Dual-luciferase reporter assay system, Promega). The screen was repeated twice and data were averaged for each luci-ferase measurement. Then, a ratio of minus beta to FL splice variants were cal-culated for each target gene. Each ratio was then expressed relative to the ratio of the siRNA control (set to a value of 1). Since many RNA-processing factors have documented effects on cell viability, we also infected cells with a SEAP reporter lentivirus. Only living cells will produce and secrete SEAP. Thus, we used SEAP as a viability control (Supplementary Figure 1B). Conditioned media (20 μL from the siRNA-transfected cells) was analyzed for SEAP (Great EscAPe SEAP Chemilu-minescence kit). As shown in Supplementary Figure 1B, siRNA depletion of *UBB* resulted in a loss of cell viability compared to the "cells only" condition. SEAP data were used following identification of a splicing hit to determine if the change in *hTERT* splicing reporter ratio was due to changes in cell viability.

**Ultraviolet immunoprecipitation RT-PCR.** UV-IP was performed as described[38] with slight modifications. Briefly, 90% confluent cells were crosslinked with UV-C (254 nm, 250 mj cm$^{-2}$), scrapped, washed, and pelleted. Pellets were then lysed in RIPA buffer containing an RNase inhibitor (Ambion), mixed with antibodies conjugated to magnetic beads (rabbit IgG or rabbit monoclonal *NOVA1*, Abcam, EPR13847, ab183024), and washed. RNA was extracted and RT-ddPCR performed with a series of primers designed near and around the in silico-identified potential *NOVA1*-binding sites (Fig. 4a, b).

**Crosslinking immunoprecipitation RT-ddPCR.** To perform CLIP we UV (UV-C, 254 nm, 250 mj cm$^{-2}$)-crosslinked cells in 15 cm dishes with 3 mL 1× PBS. A volume of 7 mL of ice-cold 1× PBS was added to the crosslinked cells and the cells were scrapped and collected by centrifugation (4 °C, $0.2 \times g$ for 5 min). Supernatant was removed and 1 mL of ice-cold 1× PBS was added and the cells spun and pelleted a second time in a 1.5 mL microcentrifuge tube. The cells were lysed in 0.5% SDS lysis buffer (Tris-HCl, pH 7, 1 mM EDTA, 1 mM dithiothreitol, RNase inhibitor (Ambion), and 1 mM phenylmethylsulfonyl fluoride; 140 μL per sample) and heated to 65 °C for 5 min and immediately placed on ice for 5 min. The lysate was volume corrected to 700 μL in RIPA correction buffer (1.25% NP40, 0.625% sodium deoxycholate, 62.5 mM Tris-HCl, pH 8, 2.25 mM EDTA, 187.5 mM NaCl, RNase inhibitor (Ambion), and 1 mM phenylmethylsulfonyl fluoride; 560 μL per sample). The lysate (700 μL) was then passed over a Qiashredder® column twice (centrifuged for 30 s at 13 000 × g at 4 °C). The entire lysate was then centrifuged for 15 min at 16 000 × g at 4 °C and the supernatant transferred to a new microcentrifuge tube. The lysates were then treated with micrococcal nuclease (0.15 U in 50 mM CaCl$_2$ buffer for 10 min at 37 °C), immediately placed on ice, and 20 mM EGTA added to quench the micrococcal nuclease activity. Antibodies and beads were prepared at room temperature in sodium phosphate buffer (0.1 M, pH 8.1; Anti-Nova1 antibody [EPR13847] (ab183024), Abcam; Rabbit IgG) and mixed with magnetic protein A/G beads (Dynabeads, 50 μL per sample) and added to the cleared lysate. The samples (NOVA1 or IgG) were immunoprecipitated at 4 °C for 4 h. The samples were then washed five times in RIPA buffer (1% NP40, 0.5% sodium deoxycholate, 0.1% SDS, 150 mM NaCl, Tris-HCl, pH 8, and 2 mM EDTA). After the final wash the bead-protein-RNA complexes were re-suspended in 200 μL of RIPA buffer and 2 U of RNase-free DNase (Ambion) was added in 300 μL of DNase buffer and incubated for 10 min at 37 °C with gentle agitation. Beads were collected with the magnet and supernatant removed. Proteinase K buffer was added (0.5 mg mL$^{-1}$ proteinase K, 0.5% SDS, 20 mM Tris-HCl, pH 7.5, 5 mM EDTA, 8 ng of MS2 RNA, and 5 μL of MRC RNA precipitation carrier; 300 μL per sample) and incubated at 37 °C for 15 min with shaking. RNA was then precipitated with sodium acetate and phenol chloroform. Following RNA precipitation reverse transcription was performed with Superscirpt III® (Invitrogen) with random hexamers. ddPCR was performed with Evagreen®.

**RNA pull down with biotinylated RNA baits**. A plasmid was generated (TOPO TA) via PCR from a BAC containing *hTERT* (RP11-990A6, CHORI) using primers that generated a 1 kb fragment of *hTERT* intron 8 including DR8. Following integration into the TOPO TA vector, in vitro transcription was performed using the T7 promoter (Ampliscribe T7 kit, Ambion, Life technologies) following the manufacturer's instructions, including a 45 min DNase step prior to RNA pre-cipitation. RNA was isolated and biotinylated at the 3′ end (Pierce RNA 3′ end biotinylation kit). Biotinylated RNA was purified with streptavidin beads. Cell lysates were prepared following the kit instructions (Peirce Magnetic RNA-protein pull-down kit). Protein-RNA complexes were immunoblotted for NOVA1 fol-lowing pull down. To produce the smaller RNA baits, T7 promoter sequences were incorporated into the 5′ end of the forward primers of each region of interest in and surrounding *hTERT* DR8. The same procedure was followed as above to generate the RNA baits. In both cases 293FT cells were transfected with V5-tagged NOVA1 cDNA construct using lipofectamine 2000. After 48 h, triplicate samples of $10 \times 10^6$ cell were washed, typisinized, counted, pelleted, and frozen at −80 °C until analysis.

**Genome-editing and engineering methods (CRISPR/Cas9 methods)**. To delete *hTERT* DR8 we designed two guide RNAs (pre-DR8 guide—5′-ATCTGCTTGC GTTGACTCGC-3′ and post DR8: 5′-TTATTTTCGGGAAGCGGCTAT-3′) and cloned these guides into PX458 (Addgene Plasmid #48138—pSpCas9(BB)-2A-GFP). Cells were transfected using Lipofectamine 3000® following the manu-facturer's instructions for scaling up to 10 cm dish. Forty-eight hours after trans-fection were flow sorted for the top 5% green fluorescent protein (GFP)-positive cells into individual wells of a 96-well plate. After about 14 days wells with growing cells were scaled up to a six-well plate and once confluent scaled up to a 10 cm dish. Cells were collected for DNA extraction and genotype analysis of CRISPR mutation validation. To validate the deletion of the 480-base pair sequence containing DR8 of *hTERT*, we performed PCR with two different primer sets of different sizes to ensure our results were robust. We also isolated the PCR product from primer set two and performed Sanger sequencing to verify the recombination event was between two *TERT* alleles and not a different sequence in the genome.

To mutate endogenous *hTERT* we had two plasmids synthesized (SGI-DNA). The WT plasmid (WT–*hTERT* intron 8 mutant PAM in pUC-SGI) contained 2000 bases of *hTERT* intron 8 surrounding DR8 that had the PAM sequence of the post-DR8 guide above mutated from 5′-CCT to ACT. The mutant plasmid (MUT-7 × "YAAY" DR8 MUT *hTERT* mutant PAM in pUC-SGI) was identical to the WT plasmid except that we mutated all seven of the "YCAY" motifs in DR8 to "YAAY" (changing the central CA to AA is known to block *NOVA1* recognition[34]). We then co-transfected the guide RNA post DR8 and either WT or MUT plasmid in the presence or absence of NHEJ inhibitor SRC7-pyrazine. We pre-treated cells with 50 nM SCR7 for 30 min prior to transfection. Cells were transfected with Lipofectamine 3000® and 48 h later the top 5% GFP-positive cells were sorted as above. Once wells with growing cells were identified we scaled up the clones as above. We isolated DNA from the clones to validate the insertion of WT or MUT plasmids. To validate WT insertion clones, we used PCR to amplify a sequence surrounding DR8 and purified and sequenced the PCR product via Sanger sequencing. To screen and validate MUT clones, we used PCR to amplify a region around DR8 and then digested the DNA with *Bcl*I. Conveniently, the mutations in DR8 (YCAY sites 4 and 5) introduced a novel restriction enzyme site that allowed us to identify mutant clones via PCR and REN digestion (similar to RFLP analysis, Supplementary Figure 5D, E, and F). We also Sanger sequence validated the MUT clones. We were able to validate two WT DR8 clones with homozygous PAM sequence mutation, indicating that homologous recombination of the donor plasmid was successful. We were also able to validate that 5 clones had homozygous integration of mutant DR8. We also sorted a variety of controls (vector only and one control exposed to both Cas9 and mutant donor that failed to undergo cutting and homology directed (HR) repair, "failed HR control") since it is well established that clonal heterogeneity exists in tumor cells lines[41]. These clones were all validated by PCR and Sanger sequencing to have WT DR8 *hTERT* sequences (representative Sanger sequencing in Supplementary Figure 5E). This produced 15 cell lines (14 clones and the parental H1299 population) that we then followed over time for telomere length, telomerase activity, and *TERT* splicing phenotypes.

**Study approval**. All animal experiments were approved by the UTSW IACUC and conducted as per institutional guidelines.

**Statistics**. Unless otherwise noted in the Methods section, figure legend, or in the Results section, pairwise Student's *t* tests (two-sided) were used to determine sta-tistically significant differences between group means. Significant differences were accepted at a *p* value < 0.05. For analysis of microarray data, we correlated the expression data of 528 RNA-binding proteins across six cell lines (differential expression analysis (*t* tests) and false-discovery rate corrected with Benjamini-Hochberg procedure).

**Data availability**. The datasets generated during and/or analyzed during the current study are available in the GEO repository under accession number

GSE32036. All other data that support the findings of this study are available from the corresponding author upon reasonable request.

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

## Acknowledgements

We thank Drs. John Minna and Adi Gazdar for cell lines and transcriptomics. We acknowledge Henrietta Lacks and her family members for the important contributions "HeLa" cells have made to this work. This work was performed in laboratories constructed with support from NIH grant C06 RR30414. This work was supported by AG01228 from the National Institute on Aging (W.E.W. and J.W.S.) and by a T32 Training Grant CA124334 (J.W.S.). A Pathway to Independence award to ATL (K99/R00 CA197672-01A1, National Cancer Institute (NCI)). We also acknowledge the Harold Simmons NCI Designated Comprehensive Cancer Center Support Grant (CA142543), the NCI SPORE (CA070907), and the Southland Financial Corporation Distinguished Chair in Geriatric Research (J.W.S and W.E.W.).

## Author contributions

M.S.W. performed the minigene screen and analysis. A.T.L. conducted all other experiments. J.D.R., L.Y., and N.D. contributed to the RT-PCR and telomerase activity analysis. K.B. performed the bioinformatics. L.Z. helped perform the cancer growth assays and quantified the data. I.M. and L.Z. helped perform the xenograft studies. M.E.S. and L.Y. performed the RNA bait assays and other assays. T.P.L. contributed reagents and assistance for telomere length analysis. A.T.L., M.S.W., J.W.S., and W.E.W. designed the study, analyzed the data, and wrote the manuscript. All authors discussed the results and commented/edited the manuscript.

## Additional information

**Competing interests:** The authors declare no competing interests.

