## [Peer Review File · Nature Communications]

Reviewers' Comments:

Reviewer #1:

Remarks to the Author:

The manuscript by Ludlow and colleagues reports the identification of Nova1 as a regulator of hTERT splicing using a combination of elegant approaches and demonstrates that altering Nova1 affects both telomere biology and cancer growth.

I have only minor comments and questions

UBB is used as a transfection control. Is it unviable according to the SEAP viability assay in Figure S1B?

Figure 2D, why do both full length forms and minus beta forms increase with the expression of Nova1, regardless of the Nova1 knockdown?

Figure 3, the effects of Nova1 knockdown on soft agar colony formation and invasion through matrigel are more extensive in H920 than in H1299 cells. Do the authors speculate it might be due to differing levels of Nova1 in these cells?

Figure S3F, Why do Nova1 levels increase upon hTERT expression?

Figure 4E, the 5 YCAY motifs in DR6 are not sufficient for Nova1 binding. Do most Nova1 targets contain more than 5 YCAY motifs, similarly to DR8?

Results of Figure 4G are not clearly explained in text and legend. Similarly for the description of results for Figure S4C-E. It is not clear that the levels of differentially spliced GLRA2 transcripts vary depending on Nova1 levels. What is NTC?

Figure 4H, why is Nova1 binding to oligo 5, which is not in the DR8 domain?

Figure 5C, telomere length of DR8 mutant clone 5 doesn't change between early and late passage. Is this because the telomere length is so short to start? Also, this clone doesn't die despite the shorter telomeres than in DR8 mutant clone 3 and DR8 deletion clone 1 (though telomeres of these 2 clones shorten the most between early and late population doublings). Have the authors looked at the shortest telomeres by Q-FISH?

Figure S5G, why have some of the DR8 mutant and deletion clones lost not only the full length hTERT but also the minus beta B isoform?

Based on the authors results and acknowledgement, there are likely other Nova1 targets that are regulating the effects on cancer cell growth and tumorigenesis observed by the authors upon Nova1 knockdown. Perhaps a short discussion of possible targets could be included.

Reviewer #2:

None

Reviewer #3:

Remarks to the Author:

NOVA1 Regulates hTERT Splicing and Cell Growth in Non-Small Cell Lung Cancer
by Ludlow et al

Ludlow et al. identify the RNA-binding protein NOVA1 as a novel regulator of hTERT splicing in lung cancer cells. The authors integrate multiple lines of evidence to conclusively show that lung cancer cell lines show variable degrees of full-length (FL) hTERT expression which is regulated by NOVA1 protein and results in increased telomerase activity. They proceed to show that NOVA1 expression has a strong impact on the tumorigenic properties of lung cancer cells, using several in vitro assays and in vivo xenografts. Notably, they confirm this in both directions, by depleting NOVA1 in a cell line with high NOVA1 and FL hTERT expression (H1299) as well as adversely by overexpressing NOVA1 in a cell line with low NOVA1 and FL hTERT expression (Calu6). They finally map the putative binding sites of NOVA1 in the hTERT pre-mRNA and remove these by CRISPR/Cas9 deletion to validate their role in NOVA1-mediated regulation of hTERT splicing.

Overall, this is a well-performed study which convincingly demonstrates a new role for NOVA1 as a critical regulator of lung cancer tumorigenesis. Since NOVA1 expression in the healthy human body is highly restricted to specific tissues, it might offer a promising target in cancer therapy. Similarly, the role of NOVA1 in hTERT splicing and telomere maintenance is clearly established by the experiments in this manuscript. The only drawback is that the authors could not rescue the phenotype of NOVA1 knockdown by forced hTERT expression (see below), leaving it open whether the observed effects of NOVA1 knockdown are indeed mediated by changed hTERT splicing. Having said this, I think that the failure to establish this final connection does not question the great overall achievements of this study.

Major comments:

1. As indicated above, the authors find that “NOVA1 knockdown was not bypassed by forced hTERT expression in the H1299 cells with NOVA1 knockdown at population doubling 60”. As pointed out in the manuscript, this indicates that “beyond the control of telomerase activity and hTERT splicing in cancer cells, NOVA1 is independently involved in pathways related to tumorigenesis and clonogenic growth”. In line with this notion, U2OS cells also showed a decreased tumour growth upon NOVA1 depletion, even though they rely on alternative lengthening of telomeres (ALT) and hence should not require FL hTERT.

As mentioned above, this result does not put into question the strong evidence that is presented in this manuscript. However, the authors should acknowledge and discuss this finding and its potential implications in more detail in the Discussion. What would be possible explanations for this observation? In the context of this, it would also be important to get more details about the experiment, e.g. what construct was used for overexpression, does it still require splicing to form FL hTERT etc. The current description of this experiment in the Materials is rather short.

2. All legends in this manuscript need to be extended to explain what is shown in the different panels.

Just as an example: what is the meaning of CMV, B6, DR6, DR8 and BGH in Fig. 1A? or what is shown by colour scale in Fig. 1C (correlation analysis) vs. Fig. 1D (differential expression analysis)?

The authors should also revise all axis labels to ensure that they are consistent between panels and convey as much information as possible. For instance, the axis in Fig. S1J currently states the R function used to generate the plot which could be exchanged by something more readable.

For all correlation coefficients, the authors should mention in the figure or legend the type of correlation measure that was used (e.g. Figure S1G-I).

Minor comments:

1. It would be helpful to show the primer locations used to measure hTERT splicing in a schematic.
2. I could not find the Supplementary Tables.
3. For ratios (like Fluc/Rluc in Fig. 1A or S1B,C), the authors should consider to use a log scale as this shows up and downregulation regulation on comparable scales.
4. To make the text easier to follow, the authors should introduce abbreviations and methods upon first appearance in figures or text, e.g. ddPCR (p. 9), VNTR (p. 26)?
5. The proteins CLK3 & SNRPB appear twice in the heatmap in Fig. 1C. If this is not a mistake, it should be explained in the figure legend.
6. In Fig. 3A,B, the figure legend should mention how an error bar was calculated for the control measurements. As I understand, all values were calculate relative to these control which are thereby invariably set to 100%.
7. In the schematic of the reporter minigenes in Fig. 4E, it is not always clear what discriminates the plasmid versions. In addition, the schematic implies that DR regions were not deleted by substituted (by what?).
8. Since the measurements in Fig. 5D were performed across multiple clones, the authors should show standard deviations or a similar measure of between-sample variability.

Response to Reviewers

We thank the reviewers and the editors for reading the manuscript and giving us a chance to respond to the comments. We have responded to all the reviewer comments. We thank the reviewers for their helpful suggestions which have significantly improved the revised manuscript. Below we have responded point-by-point to each of the reviewers' comments in **red text**.

Reviewer #1 (Remarks to the Author):

The manuscript by Ludlow and colleagues reports the identification of Nova1 as a regulator of hTERT splicing using a combination of elegant approaches and demonstrates that altering Nova1 affects both telomere biology and cancer growth.

-We thank the reviewer for recognizing the novelty and importance of this work and have carefully considered the minor comments in our revised manuscript.

I have only minor comments and questions

UBB is used as a transfection control. Is it unviable according to the SEAP viability assay in Figure S1B?

-The SEAP viability for the UBB transfected cells was 0.08 indicating that there were very few viable cells. For example, the cell only control was set to 1.0 in the SEAP analysis. Transfection of the HeLa cells with the hTERT minigene reporter with the siRNA against UBB resulted in very few surviving cells. We clarified this point in the revised manuscript on page 7 as below: 'As shown in Supplemental Figure 1B, siRNA depletion of ubiquitin (*UBB*) resulted in a loss of cell viability compared to the 'cells only' condition.'

Figure 2D, why do both full length forms and minus beta forms increase with the expression of Nova1, regardless of the Nova1 knockdown?

- While we are not completely sure why NOVA1 overexpressed in H1299 cells results in both FL and minus beta increases, this could be due to a transcription factor being spliced or another upstream event that NOVA1 is involved in that results in increased expression (transcription) of TERT. Furthermore, recent RNA sequencing data from another group studying the role of NOVA1 in pancreatic beta cells found that NOVA1 regulated a wide variety of transcripts including several transcription factors that could be acting upstream of TERT. We have commented on this point in the revised manuscript on page 14 as follows:

'One possibility is that the increased levels of *hTERT* (both FL and minus beta observed in the rescue experiment, Figure 2D specifically) is due to a transcription factor being spliced or another upstream event that NOVA1 is involved in results in increased expression (transcription) of *hTERT*. Recent RNA sequencing data indicates that NOVA1 may regulate the expression and splicing of transcription factors that could be acting upstream of *hTERT* in cancer cells³³.'

Figure 3, the effects of Nova1 knockdown on soft agar colony formation and invasion through matrigel are more extensive in H920 than in H1299 cells. Do the authors speculate it might be due to differing levels of Nova1 in these cells?

-We agree with the reviewer that higher expression levels of NOVA1 results in a more robust impact of NOVA1 knockdown, as is the case between H1299 and H920 cells. Further evidence supporting this idea is from the cell line H2882. H2882 cells have very robust levels of NOVA1, higher than both H1299 and H920. When we knocked down NOVA1 in this cell line it only divided twice in 90 days (supplemental Figure 2s)

indicating that there appears to be a correlation between the level of NOVA1 expression and cellular response to knockdown of NOVA1. We have added a statement to this effect in the revised manuscript on page 17 as follows:

‘Interestingly, the levels or amount of NOVA1 in the cell lines (H1299 moderate NOVA1 compared to high levels of NOVA1 in H920 cells) seems to correlate the response of the cells to knockdown of NOVA1. For instance, the impact of NOVA1 knockdown on colony formation and migration was much greater in H920 cells compared to H1299 cells.’

Figure S3F, Why do Nova1 levels increase upon hTERT expression?

- This observation could be a hypermorphic phenotype where TERT is acting in a way it typically would not due to the extremely high levels of TERT being expressed from the CMV driven promoter of the cDNA plasmid. Alternatively, there could be a biological regulatory loop where high TERT drives increased expression of NOVA1 as some sort of cell growth promoting phenotype. Additional experiments, well beyond the scope of this manuscript, would be needed to provide evidence for either possible explanation. However, in the revised manuscript we have added a short discussion on this issue on page 18 as follows:

‘Interestingly, upon *hTERT* overexpression the levels of NOVA1 increased in both the shRNA control and shRNA to NOVA1 lines. This observation can indicate at least two possibilities; either a hypermorphic phenotype where *TERT* is acting in a way it normally would not due to the high expression levels or that a regulatory loop exists between NOVA1 and *hTERT*.’

Figure 4E, the 5 YCAY motifs in DR6 are not sufficient for Nova1 binding. Do most Nova1 targets contain more than 5 YCAY motifs, similarly to DR8?

-Yes, most NOVA1 target genes contain clusters of YCAY motifs. The specificity of why NOVA1 binds certain YCAY motifs over others were not identified in our present studies. However, Robert Darnell’s group at NYU has studied NOVA1’s motif extensively and still does not have a good model for which YCAY clusters/elements NOVA1 prefers over others. This is an active and exciting part of current RNA biology research. Further, the specificity of NOVA1 for certain YCAY elements over others could be dictated by additional splicing factors that cooperate to influence splicing. In terms of binding partners in the DR8 region we are actively investigating this in our laboratory and trying to determine what else may be acting in combination with NOVA1 at DR8 that helps drive the specificity in addition to the enhancement or repression of full-length transcripts. There are likely other sequences in and around DR8 that attract additional splicing factors that in turn determine NOVA1’s impact on TERT splicing.

Results of Figure 4G are not clearly explained in text and legend. Similarly, for the description of results for Figure S4C-E. It is not clear that the levels of differentially spliced GLRA2 transcripts vary depending on Nova1 levels. What is NTC?

- We apologize for the oversights and have clarified our discussion of Figure 4G and have expanded the legend in our revised manuscript. We point the reviewer to Supplemental Figure 4D in H1299 cells. When NOVA1 levels are reduced, the use of

GLRA exon 3B is reduced and the levels of GLRA exon 3A go up moderately, which is reversed by NOVA1 rescue. In supplemental Figure 4C, we are just pointing out that GLRA 3B is highly expressed in HeLa cells while GLRA exon 3A is low expressed or not expressed at all. We have clarified this in the results section. Further, the point of Supplemental Figure 4C-E is to show that a known target of NOVA1 is able to be assayed in our cDNAs showing that our CLIP technique is robust. We have clarified this point in the results sections as suggested.

We modified the text in the revised manuscript on page 21 as follows:

‘As previously described *GLRA2* exon 3A and 3B are mutually exclusive exons regulated by NOVA1. First, we tested for expression of *GLRA2* mutually exclusive exon 3A and 3B usage in HeLa and H1299 cancer cells (Supplemental Figure 4C and E), and observed that exon 3B of *GLRA2* was preferentially used. Next, we looked in our H1299 *NOVA1* rescue series to see if *GLRA2* was regulated by NOVA1 in cancer cells and indeed found that *GLRA2* exon 3B is preferentially used when NOVA1 levels are higher (Supplemental Figure 4D) similar to previous studies⁴¹. Next, we assayed our CLIP cDNAs and observed that *GLRA2* was effectively pulled down in all extracts regardless of *hTERT* status, indicating that our CLIP was efficient (Supplemental Figure 4E).’

NTC = no template control and we have added this statement to the legend as well.

Figure 4H, why is Nova1 binding to oligo 5, which is not in the DR8 domain?

-To clarify, in oligo 5 there are 2 YCAY motifs. NOVA1 may be directly binding these two YCAY motifs or could be pulled down by other factors interacting with this oligo as part of a complex. We speculate that oligo 5, which is 71 nucleotides 3' of DR8 and is 127 nucleotides long may be bound by proteins that are interacting with NOVA1 in a splicing factor complex. Thus, when the western blot is performed, NOVA1 is observed at reduced levels compared to direct interactions as in oligo 3 (4 YCAY motifs) and oligo 4 (5 YCAY motifs). We have added a statement in the revised manuscript on page 22 about this issue.

‘We hypothesize that NOVA1 is binding to DR8 and the region surrounding DR8 of *hTERT* through both direct and indirect interactions (that is, as part of a splicing factor complex) that directs NOVA1's impact on *hTERT* splicing. There are 7 YCAY motifs in DR8 (oligos 3 and 4), oligo 5, which is 71 nucleotides 3' of DR8, contains an additional 2 YCAY motifs. We hypothesize that a complex of proteins, including NOVA1, that interact at or near DR8 could be pulling down NOVA1 with oligo 5, explaining why NOVA1 is present in oligo 5's pulldown.’

Figure 5C, telomere length of DR8 mutant clone 5 doesn't change between early and late passage. Is this because the telomere length is so short to start? Also, this clone doesn't die despite the shorter telomeres than in DR8 mutant clone 3 and DR8 deletion clone 1 (though telomeres of these 2 clones shorten the most between early and late population doublings). Have the authors looked at the shortest telomeres by Q-FISH?

-We thank the reviewer for this comment and think they are referring to signal free ends as determined by Q-FISH. We have not performed this assay on the hTERT DR8

mutant or deletion cells. Further, we think the reviewer is referring the DR8 mutant clone 4 which has extremely short telomeres that appear to have the lowest shortening rate of any of the mutant clones. One possibility is that the short telomeres in these clones are being maintained by the small amount of FL TERT at the population doublings we analyzed. Also, from the literature it is likely that telomere length heterogeneity exists between cancers types and within specific tumor cell lines. Also, telomerase inhibition therapy has pointed out that some cell lines die when average telomere length are at 2 kb while others die at 5kb. The mechanisms of these observations are not well understood. We speculate that these differential responses have to do with how DNA damage is being sensed in these cells, specifically telomere end specific DNA damage. For example, clones or cells that are very good at sensing telomere end DNA damage would die before cells that suppress or are poor at detecting telomere end DNA damage. In addition, telomerase recruitment and processivity may have a role in how very low levels of telomerase can maintain very short telomeres in cancers (i.e., what is occurring in Clone 4 in the current manuscript). This series of clones could offer insights into these other questions that are beyond the scope of the current manuscript. However, this interesting point is now described in the discussion section of our revised manuscript on pages 27-28.

‘One interesting aspect of this study was the generation of CRISPR/Cas9 clones of H1299 lung cancer cells. We observed the expected telomere length heterogeneity in the wild-type clones, however some striking and unexpected phenotypes arose upon mutation and deletion of DR8 in hTERT. For instance, the two clones that eventually died off in culture had about 2 kb telomere lengths when they died (DR8 mutant clone 3 and DR8 deletion clone 1). However, DR8 mutant clone 4 had a mean telomere length of about 2 kb and survived in culture (Figure 5). We speculate that the telomere DNA damage sensing capabilities between the clones that died compared to the clones that survived at these short telomere lengths are different. Further, DR8 mutant clone 3 and DR8 deletion clone 1 both lacked expression of hTERT exons 6-9, while DR8 mutant clone 4 had a small amount of FL hTERT remaining (Figure 5). This small amount of hTERT FL could have generated enough active telomerase to maintain the short telomeres. Further, we speculate DR8 mutant clone 4 could have had recruitment and processivity advantages compared to other lines with higher telomerase activity but increased rates of telomere shortening. These clones could be useful in future studies aiming at elucidating the genetic and molecular underpinnings of these less well described phenomena.’

Figure S5G, why have some of the DR8 mutant and deletion clones lost not only the full-length hTERT but also the minus beta B isoform?

- This is an interesting observation. One possibility is that disruption of DR8 (whether mutation or deletion) in certain clones may differentially express other splicing factors beyond NOVA1 that in turn may be regulating TERT splicing at DR8. The DR8 mutations and deletions could also be changing RNA secondary structure. Changes in RNA secondary structures are well documented to alter splicing. We point the reviewer to the observation that all of the clones still have TERT chromosomal DNA and also that TERT is still expressed (Supplemental Figures 5C, F, and H) indicating that it is most

likely post-transcriptional mechanisms defining what TERT variants are present. A short discussion of this is included in the revised manuscript on page 29.

'Supporting this model, when DR8 is deleted or all 7 YCAY motifs in DR8 are mutated three lines switched their splicing to almost solely minus beta (Supplemental Figure 5). In one DR8 mutant (clone 3) and one DR8 deletion (clone 1) we observed a complete loss of expression of *hTERT* exons 6-9 (supplemental Figure 5). One possibility is that the loss of NOVA1 binding in these clones leads to a change in secondary structure in the pre-mRNA of *hTERT* and results in a complete skipping of exons 6-9 but that other regions of *hTERT* are still expressed (Supplemental Figure 5H).'

Based on the authors results and acknowledgement, there are likely other Nova1 targets that are regulating the effects on cancer cell growth and tumorigenesis observed by the authors upon Nova1 knockdown. Perhaps a short discussion of possible targets could be included.

-We agree and we have included a short discussion of potential targets of NOVA1 that could be regulating the growth of cancer cells.

The following discussion has been added to the manuscript as requested on pages 30-31:

"From our data and that of others, it is clear that NOVA1 has additional targets in cancer cells that are related to the growth phenotypes observed herein. Other potential targets of NOVA1 that could be influencing growth and survival of cancer cells include the observation that NOVA1 may regulate AKT levels. The role of AKT in cell proliferation and survival is well documented and several anti-cancer drugs are currently in use that target AKT. Further, AKT is upstream of FOXO transcription factors which control cell survival and apoptosis. AKT functions in cellular survival by phosphorylating and sequestering FOXO's in the cytoplasm and keeping them inactive and unable to initiate transcription of the cell death program. When NOVA1 levels are reduced, AKT protein levels are reduced, FOXO's become active and initiate transcription of the cellular death machinery. Future studies investigating the role of NOVA1 in AKT induced cancer cell survival are warranted. Further, AKT is also a kinase that may increase the nuclear pool of TERT proteins and thus increase the potential for active telomere length maintaining telomerase. Thus, a NOVA1-AKT-TERT axis may exist in some cancers that may be targetable."

Reviewer #2 (Remarks to the Author):

NOVA1 Regulates hTERT Splicing and Cell Growth in Non-Small Cell Lung Cancer
by Ludlow et al

Ludlow et al. identify the RNA-binding protein NOVA1 as a novel regulator of hTERT splicing in lung cancer cells. The authors integrate multiple lines of evidence to conclusively show that lung cancer cell lines show variable degrees of full-length (FL) hTERT expression which is regulated by NOVA1 protein and results in increased telomerase activity. They proceed to show that NOVA1 expression has a strong impact on the tumorigenic properties of lung cancer cells, using several in vitro assays and in vivo xenografts. Notably, they confirm this in both directions, by

depleting NOVA1 in a cell line with high NOVA1 and FL hTERT expression (H1299) as well as adversely by overexpressing NOVA1 in a cell line with low NOVA1 and FL hTERT expression (Calu6). They finally map the putative binding sites of NOVA1 in the hTERT pre-mRNA and remove these by CRISPR/Cas9 deletion to validate their role in NOVA1-mediated regulation of hTERT splicing.

Overall, this is a well-performed study which convincingly demonstrates a new role for NOVA1 as a critical regulator of lung cancer tumorigenesis. Since NOVA1 expression in the healthy human body is highly restricted to specific tissues, it might offer a promising target in cancer therapy. Similarly, the role of NOVA1 in hTERT splicing and telomere maintenance is clearly established by the experiments in this manuscript. The only drawback is that the authors could not rescue the phenotype of NOVA1 knockdown by forced hTERT expression (see below), leaving it open whether the observed effects of NOVA1 knockdown are indeed mediated by changed hTERT splicing. Having said this, I think that the failure to establish this final connection does not question the great overall achievements of this study.

We thank the reviewer for indicating our studies were well-performed and convincingly demonstrating a new role for NOVA1 in lung cancer. We have addressed the other comments below.

Major comments:

1. As indicated above, the authors find that “NOVA1 knockdown was not bypassed by forced hTERT expression in the H1299 cells with NOVA1 knockdown at population doubling 60”. As pointed out in the manuscript, this indicates that “beyond the control of telomerase activity and hTERT splicing in cancer cells, NOVA1 is independently involved in pathways related to tumorigenesis and clonogenic growth”. In line with this notion, U2OS cells also showed a decreased tumour growth upon NOVA1 depletion, even though they rely on alternative lengthening of telomeres (ALT) and hence should not require FL hTERT.

As mentioned above, this result does not put into question the strong evidence that is presented in this manuscript. However, the authors should acknowledge and discuss this finding and its potential implications in more detail in the Discussion. What would be possible explanations for this observation? In the context of this, it would also be important to get more details about the experiment, e.g. what construct was used for overexpression, does it still require splicing to form FL hTERT etc. The current description of this experiment in the Materials is rather short.

-We thank the reviewer for this comment and we have expanded the discussion of the failure of hTERT forced expression to rescue the NOVA1 induced cell growth phenotypes and of the ALT cell experiments. We can only speculate that beyond hTERT, NOVA1 regulates a variety of transcripts important for growth and survival. For instance, a recent paper studying NOVA1 in pancreatic beta cells observed that NOVA1 regulates AKT and FOXO transcription factors, both of which are important growth and survival pathways. Thus, NOVA1 in ALT cells may be regulating growth and survival by manipulating AKT and FOXO, providing a potential explanation for the observed results. A short discussion similar to the above paragraph has been added to the manuscript as requested on pages 30-31.

- In terms of the hTERT rescue experiment we point the reviewer to the results section on page 17. In the methods section the following text on page 33 was added:
“To test the dependence of observed growth defects on telomere length and telomerase activity, an *hTERT* cDNA (pMIN-Ub-IRES-Blast lentiviral vector from (37)) was transfected into H1299 cells with control shRNAs or NOVA1 shRNA after 50 population doublings. This vector contains a 3.398 kb cDNA of *hTERT* coding sequence thus no splicing is needed to make hTERT mRNA. Further information about the hTERT cDNA from pGRN145 that was inserted into a lentiviral vector can be found here (<https://www.atcc.org/Products/All/MBA-141.aspx#generalinformation>). We convincingly showed that the TERT RT domain was expressed in our experiments and that telomerase was active (see Supplemental Figure 5). Following infection and selection, cells were plated as described above in the cancer growth assays.”

2. All legends in this manuscript need to be extended to explain what is shown in the different panels.

- We have expanded the legends to provide more precise detail of what is shown in each panel.

Just as an example: what is the meaning of CMV, B6, DR6, DR8 and BGH in Fig. 1A? or what is shown by colour scale in Fig. 1C (correlation analysis) vs. Fig. 1D (differential expression analysis)?

- We have added labels and descriptors in the legend for the heatmap scales.

The authors should also revise all axis labels to ensure that they are consistent between panels and convey as much information as possible. For instance, the axis in Fig. S1J currently states the R function used to generate the plot which could be exchanged by something more readable.

- We have revised the axis labels to be consistent and have updated the labels with better descriptors.

For all correlation coefficients, the authors should mention in the figure or legend the type of correlation measure that was used (e.g. Figure S1G-I).

- We have added in labels and described in the legends what correlation analysis was used in each plot.

Minor comments:

1. It would be helpful to show the primer locations used to measure hTERT splicing in a schematic.

- We have included a cartoon to go with the supplemental table 2 of the primer locations for hTERT. This information and the new cartoon can be found in Supplemental Table 2.

2. I could not find the Supplementary Tables.

- We apologize that the reviewer could not find the supplemental tables. We ensured they were uploaded in the first round and have also made sure they are a part of this revised version as well.

3. For ratios (like FLuc/RLuc in Fig. 1A or S1B,C), the authors should consider to use a log scale as this shows up and downregulation regulation on comparable scales.

- We thank the reviewer for this suggestion. We have opted to replace the former graph in Figure 1 in favor of the suggested log₂ scale graph.

4. To make the text easier to follow, the authors should introduce abbreviations and methods upon first appearance in figures or text, e.g. ddPCR (p. 9), VNTR (p. 26)?

- We have revised the text to describe each abbreviation upon first use.

5. The proteins CLK3 & SNRPB appear twice in the heatmap in Fig. 1C. If this is not a mistake, it should be explained in the figure legend.

- This is not a mistake. In the microarray data these genes have multiple probes that significantly correlated with the hTERT FL measures. Thus, they are included twice. This fact is now mentioned in the legend of the revised manuscript to clarify the reviewer's point.

6. In Fig. 3A,B, the figure legend should mention how an error bar was calculated for the control measurements. As I understand, all values were calculate relative to these control which are thereby invariably set to 100%.

- We have added the following text to the legend of figure 3 explaining how we calculated a measure of variance around the control samples.

'#NOTE – An error bar on the controls of Figure 3 A, B and C was calculated by first generating the mean of the controls and expressing each control relative to this value. We then calculated a standard deviation and standard error based on the variance observed between these normalized values.'

7. In the schematic of the reporter minigenes in Fig. 4E, it is not always clear what discriminates the plasmid versions. In addition, the schematic implies that DR regions were not deleted by substituted (by what?).

- We apologize for the lack of clarity and thank the reviewer for pointing this out. These are indeed deletions. We have modified the figure to clearly show that the DNA was completely removed resulting in plasmids of different lengths.

8. Since the measurements in Fig. 5D were performed across multiple clones, the authors should show standard deviations or a similar measure of between-sample variability.

- We have added in a new figure showing the variability between FL and spliced products. The new figure is Fig. 5E. All subsequent figure panels in Figure 5 are re-labeled and updated in the text.

Reviewers' Comments:

Reviewer #1:

Remarks to the Author:

The authors have properly addressed all comments.

Reviewer #2:

Remarks to the Author:

All my comments have been addressed.